# Molecular basis for multidrug efflux by an anaerobic-associated RND transporter

Ryan Lawrence[1,2], Mohd Athar [3], Muhammad R. Uddin[4], Christopher Adams[5], Joana S. Sousa[5], Oliver Durrant [5], Sophie Lellman[5], Lucy Sutton[1], C. William Keevil [1], Nisha Patel[5], Christine E. Prosser[5], David McMillan [5], Helen I. Zgurskaya [4], Attilio V. Vargiu [3], Zainab Ahdash [5] ✉ & Eamonn Reading [1,2] ✉

Bacteria can resist antibiotics and toxic substances within demanding ecological settings, such as low oxygen, extreme acid, and during nutrient starvation. MdtEF, a proton motive force-driven efflux pump from the resistance-nodulation-cell division (RND) superfamily, is upregulated in these conditions but its molecular mechanism is unknown. Here, we report cryo-electron microscopy structures of *Escherichia coli* multidrug transporter MdtF within native-lipid nanodiscs, including a single-point mutant with an altered multidrug phenotype and associated substrate-bound form. Drug binding domain and channel conformational plasticity likely governs substrate polyspecificity, analogous to closely related, constitutively expressed counterpart, AcrB. Whereas we discover distinct transmembrane state transitions within MdtF, which create a more engaged proton relay network, altered drug transport allostery and an acid-responsive increase in efflux efficiency. Our findings provide mechanistic insights necessary to understand bacterial xenobiotic and toxin removal by MdtF and its role within nutrient-depleted and acid stress settings, as endured in the gastrointestinal tract.

Presenting as an increasing global health challenge that demands immediate intervention, bacterial antimicrobial resistance (AMR) was attributed to approximately 1.14 million deaths in 2021[1]. Bacteria can acquire resistance through multiple mechanisms, however, one of the most pertinent is the overexpression of efflux pumps. The Resistance-Nodulation-Division (RND) family of efflux transporters comprises the most clinically relevant efflux pumps in Gram-negative bacteria[2,3]. These large protein conduits span their double membrane as a tripartite assembly comprising an inner-membrane RND transporter, an outer-membrane channel, and a periplasmic membrane fusion protein which connects the other two (also known as periplasmic adaptor proteins), forming a tunnel through which substrates are translocated to the external environment[4,5] (Fig. 1A). Energised by the proton

motive force (PMF), the polytopic RND proteins export an extensive range of chemically and structurally dissimilar antibiotics, thereby conferring multidrug resistance (MDR) in clinical isolates[4].

*Escherichia coli* (*E. coli*) and closely related *Shigella* species are facultative anaerobes that can cause severe infections in humans and exhibit varying susceptibilities to antibiotic treatments dependent on the oxygen levels within the environment they occupy[6]. Here, the constitutive AcrAB-TolC efflux pump from *E. coli* provides intrinsic resistance and remains the archetypal example of RND pumps. Extensive research on this system has facilitated a more precise structural comprehension of the mechanism of efflux adopted by this family[7–9]. Structural and biochemical studies have revealed that the homotrimeric AcrB transporter undergoes a functional rotation

[1]School of Biological Sciences, University of Southampton, Southampton, UK. [2]Department of Chemistry, Britannia House, King's College London, London, UK. [3]Department of Physics, University of Cagliari, Cittadella Universitaria, S.P. Monserrato-Sestu, Monserrato, CA, Italy. [4]Department of Chemistry and Biochemistry, University of Oklahoma, Norman, OK, USA. [5]Department of Protein Structure and Biophysics, UCB Biopharma, Slough, UK. ✉e-mail: zainab.ahdash@ucb.com; e.reading@soton.ac.uk

mechanism to translocate its substrates, going through sequential cycling of asymmetric protomer conformations: access, binding, and extrusion. This conformational process being simultaneously powered by transmembrane proton-relay from the periplasm to the cytoplasm. Substrates enter binding pockets housed within the porter domain through a series of substrate-selective channels and pathways identified in an access state. During transition to a binding state substrates are brought towards the distal binding pocket (DBP) following conformational changes in the porter domain[4,10–12]. The DBP is separated from a proximal binding pocket (PBP) by a switch loop, which is necessary to regulate substrate binding and export due to its intrinsic flexibility[13,14]. Upon further conformational change to an extrusion state, substrates are then transported towards the exit tunnel at the top of the periplasmic domain of AcrB, which extends through AcrA and into TolC. The extrusion protomer then returns to the access state upon release of the substrate, restarting the functional rotation mechanism whereby the homotrimer can undergo another cyclic event[10,11]. Site-directed mutagenesis studies revealed that four transmembrane domain residues, D407, D408, K938, and R969, are essential to the underlying energy transduction during this conformational switching[15,16]. However, the exact mechanism of coupling proton and substrate antiport for RND multidrug transporters is still an ongoing area of research, even for the prototypical AcrB.

As just described, most of our knowledge on RND-based efflux comes from *E. coli* AcrB, however, little is known about the molecular mechanism of its closely related homologue, MdtF. MdtF forms part of the MdtEF-TolC efflux pump (formerly known as YhiUV-TolC[17]), which can secrete a similarly wide range of chemically diverse substrates as AcrB, including antibiotics. Markedly, *mdtEF* is dramatically upregulated during anaerobic conditions (~20-fold increase in expression), independent of antibiotic exposure, compared to all twenty *E. coli* K-12 efflux genes and provides enhanced drug tolerance in anaerobically grown *E. coli* (Fig. 1A)—being linked to the expulsion of host-derived toxic nitrosyl indole derivatives as bacteria respire nitrate instead of oxygen[18]. *mdtEF* is also upregulated in other nutrient-deprived and environmental stress conditions, such as during growth-cessation (with a 14/41-fold increased expression at early/late stationary phases), biofilm formation, low pH, iron starvation, and during catabolite induction[18–24]. Coincidentally, *E. coli* cell envelope remodelling forms a mutual role in bacterial protection by reducing cell wall permeability, allowing bacterial cells to survive in changing settings such as hypoxic stress and stationary phase progression[25]. As MdtF is an inner membrane transporter, then it is worth noting that, as part of this response, the inner membrane is altered whereby unsaturated fatty acids of the inner membrane are converted to cyclopropane fatty acids. Here, a methylene group, deriving from *S*-adenosyl-L-methionine (SAM) is transferred to an unsaturated fatty acid chain in a reaction catalysed by cyclopropane fatty acid synthase. This enrichment of cyclopropanated lipids exerts protective effects against extreme temperatures, low pH, and other environmental perturbations by decreasing permeability and increasing inner membrane rigidity[26–28]. Although these pertinent mechanisms highlight the potential clinical and physiological significance of the MdtEF-TolC pump, molecular details which describe the multidrug recognition and translocation by the MdtF inner membrane transporter is lacking.

In this work, we report three cryogenic-electron microscopy (cryo-EM) structures of MdtF, including substrate-free (apo-MdtF$^{WT}$ and apo-MdtF$^{V610F}$) and substrate-bound (R6G-MdtF$^{V610F}$) states. The V610F mutation in MdtF, located in the DBP, was previously identified during antibiotic selective pressure (under aerobic culturing) and engendered increased susceptibility to macrolides, but increased resistance to chloramphenicol, fluoroquinolones, linezolid, and tetracycline[29]. In combination with molecular dynamics and functional assays, we uncover that MdtF powers its functional rotation through distinct transmembrane conformational changes,

which delivers moderate efflux under neutral conditions, but an acid-responsive increase in efflux efficiency. This duality has not yet been found in other RND efflux pumps and may be important when considering pump energy expense and fitness costs—of MdtF and other RND transporters—within the transitory anaerobic and acidic microhabitats found in the mammalian gut, where *E. coli* and *Shigella* bacteria colonise and infect. Furthermore, we also reveal that the single-point mutation V610F causes swelling of the DBP and changes porter domain arrangements, which likely determines its altered multidrug efflux profile. Our results add to growing evidence that conformer allostery are the determinants of RND efflux pump function and specificity, and that even single-point mutations can drastically influence multidrug resistance phenotypes by altering conformational plasticity[30].

## Results

### Cryo-EM structure of MdtF unveils its functional rotation mechanism

To obtain structures in a 'native' lipid environment, MdtF was overexpressed in the *E. coli* C43Δ*acrAB*(DE3) strain and extracted in SMA lipid particles (SMALPs) from stationary phase bacterial membrane fractions[31–33]. These polymer nanodiscs encapsulate membrane proteins within the intrinsic lipid mix of cells and are assembled via spontaneous portioning of the polymer into the bilayer. Native nanoparticles of *E. coli* MdtF$^{WT}$ were purified to homogeneity before cryo-EM structural determination (Supplementary Fig. 1). The structure of *E. coli* MdtF$^{WT}$ was then determined by single-particle cryo-EM to a global resolution of 3.56 Å (Fig. 1B). Cryo-EM data processing and analysis, and an assessment of cryo-EM map quality is delineated in Supplementary Figs. 2–7 and Table 1. To avoid the loss of lipid-bilayer structural information, the map reconstruction was performed with C1 symmetry. Here, we observe density for several lipids encompassing and buttressing the transmembrane region of MdtF in our structures, with those occupying the central cavity of the transmembrane region being better resolved (Fig. 1C). In the apo-MdtF$^{WT}$ structure, we were able to resolve 21 lipid molecules and an additional 7 alkyl chains molecules within the central cavity. Phosphatidylethanolamine (PE) was chosen as the ligand to fit in the elongated density of our structural model, as lipid analysis revealed it to be predominant within the nanodisc sample, with small amounts of phosphatidylglycerol (PG) and cardiolipin (CL) also detected. The phospholipid fatty acid content of the nanodisc was determined by GC-MS to comprise of ~ 50%, 26% and 24% unsaturated, saturated and cyclopropanated (UFA, SFA, and CFA), respectively (Supplementary Fig. 8 and Supplementary Table 2). The resolved lipid bilayer is approximately 30 Å thick and exists as two triangular bilayer leaflet arrangements rotated relative to one another, with the inner leaflet being somewhat more organised than the outer leaflet chains, a phenomenon also found in AcrB[31]. Due to proximity and lack of regular arrangement between each monomer, this could indicate the importance of the lipid region in MdtF functionality and further highlights the amenability of this native purification system in membrane protein-based study.

Our resulting structural model of MdtF showed the expected analogous domains and loops shared amongst RND transporters (Fig. 1D) and possessed asymmetry between individual protomers (Fig. 1E, and Supplementary Fig. 10A); this functional asymmetry was not captured by AlphaFold[34] predicted structures (Supplementary Fig. 11), with accurate asymmetrical structural prediction a known challenge for the method[35]. Comparison to published structures of RND transporters AdeB[36], AcrD[37], AcrB[31,38], OqxB[39], MexB[40] and CusA[41] revealed that MdtF is most like AcrB (Supplementary Table 7). This is, perhaps, expected as MdtF is part of the same phylogenetic cluster as AcrB and shares high substrate profile[42] and sequence similarity (~ 71%, Supplementary Fig. 9), MdtF being distinct rather in its dramatic upregulation in anaerobic and growth-sessile conditions. Therefore,

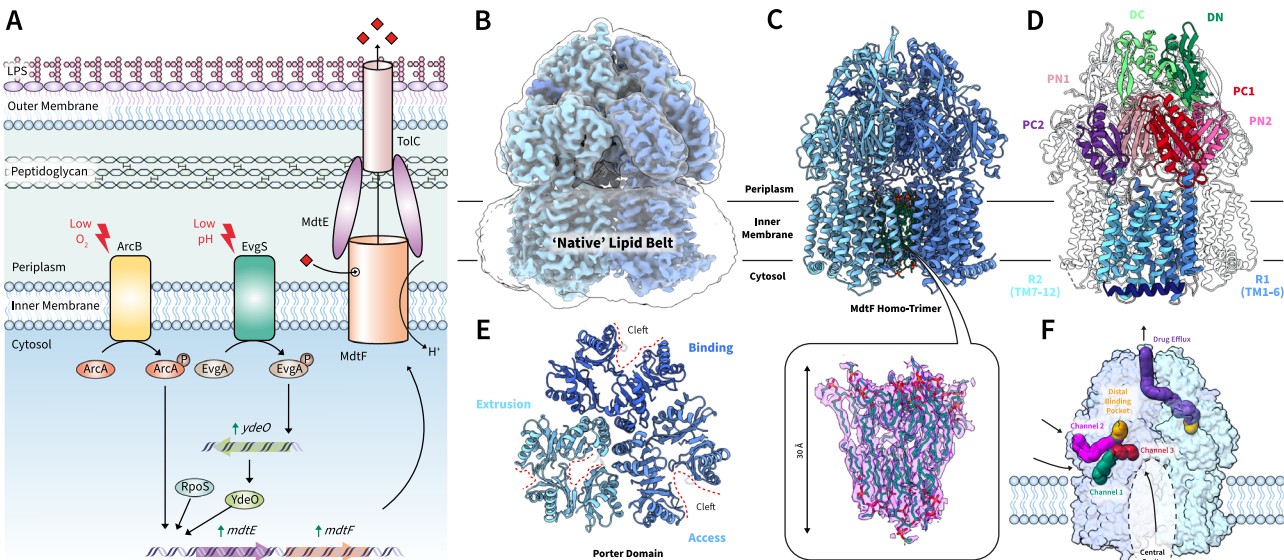

**Fig. 1 | MdtF regulation and structural features. A** *mdtEF* expression is regulated by the two-component system EvgAS, a signal transduction system that confers acid resistance to *E. coli*. *mdtEF* expression was also purported to be controlled by the conserved alternative sigma factor RpoS, required for *E. coli* survival in stress conditions associated with starvation, such as osmotic challenges, heat shock, and oxidative stress. In addition, the upregulation of *mdtEF*, accompanied by increased efflux activity and drug tolerance, is observed in anaerobically grown *E. coli*. During this condition, the upregulation of *mdtEF* is induced by the global anaerobic regulatory factor ArcA, which controls the expression of genes related to the shift in the mode of energy metabolism from aerobic to anaerobic. **B** Structure of apo-MdtF[WT] purified in SMALPs solved by cryo-EM to a resolution of 3.56 Å, demonstrating the maintenance of the native lipid belt encompassing the transmembrane domain of the protein in the electron density map. **C** Ordered lipids buttressing the transmembrane domain of the three protomers of MdtF can be observed as elongated density in the cryo-EM structure. These were modelled in as PE lipids. **D** Domains and subdomains of MdtF. **E** Top view of the periplasmic porter domain, demonstrating its asymmetric structure with open clefts observed within the access and binding states. **F** Putative substrate translocation channels of MdtF[WT] as calculated by MOLE[106].

we posit that a focused comparison with AcrB is necessary to understand the structural underpinnings of MdtF function.

Global assignment of protomers in accordance with AcrB-related asymmetry revealed similar molecular dissimilarities between the 'chain B' protomer of MdtF with classified access and binding states in AcrB; a finding consistent when comparing to AcrB structural models solved in either DDM detergent micelle or SMALP membrane mimetic environments ($C_\alpha$ RMSD's of 1.6/1.9 Å and 2.3/2.0 Å against AcrB access/binding states in DDM and SMALPs, respectively; Fig. 2A and Supplementary Table 7). Whereas AcrB binding and extrusion states match well with respective MdtF chains 'C' and 'A' across both mimetic environments (< 1.5 Å global RMSD of $C_\alpha$'s). This made the access state hard to discern and was suggestive of a distinct access state architecture in-between access and binding states. Indeed, the structure of the MdtF access protomer does not correspond well to any other characterised RND transporter protomer (> 1.8 Å global RMSD of $C_\alpha$'s) (Supplementary Table 7). Next, we investigated local conformational RMSD ($C_\alpha$-atom) variance between the asymmetric trimer states of MdtF and AcrB. The funnel and porter domains remain largely, structurally conserved across all three protomers creating similar putative substrate translocation channels to AcrB (Fig. 1F and Supplementary Fig. 10B, C). While there is discernible conformational discrepancy within a portion of the extrusion protomer that forms the substrate exit pathway (Fig. 2B), it is within the transmembrane domains where the structures begin to deviate most prominently, which we describe below.

Strikingly, within our MdtF structure, the identified access state already possesses a binding-like R2 state whereby it appears to be structurally 'swung out' (Fig. 2C), making the R2 region positioning near-identical between access and binding states. This is also corroborated by the lack of conformational translocation of the Iα helix during the transition between access and binding states. This contrasts with AcrB, where distinct R2 conformers are observed between

access and binding states. The significant R2 (access) variance between the two homologues, AcrB and MdtF, is consistent in both detergent and SMALP environments (Fig. 2B, C and Supplementary Fig. 12). Moreover, in MdtF[WT], the TM2 helix seems to remain 'stationary' through its binding to extrusion transition, moving only ~0.3 Å in the z-direction of the membrane plane compared to ~ 2.5 Å in AcrB (Fig. 2C and Supplementary Fig. 13A). Whereas AcrB-like alterations in the hoisting loop from flexible to helical in structure remain within the asymmetric functional rotation of MdtF (Supplementary Fig. 13B).

It is important to consider these transmembrane configurations in MdtF alongside the current interpretation of their positions within the functional rotation mechanism of the prototypical RND transporter, AcrB. AcrB transition between access and binding states leads to PC1 and PN2 subdomains undergoing a structural change that causes the entire R2 domain (TM7-12) to swing away due to a pushing motion placed on the Iα helix arising from the TM2 downward displacement. The transmembrane domain is flanked by two transmembrane helices, TM2 and TM8, which provide passive transmission of conformational energy to the porter domain, and vice versa, in AcrB[16,43]. Where the C-terminal end of TM8 harbours the essential 'hoisting loop' which confers its structural movements to the porter domain, coupling its dynamics to allosteric transmembrane domain transitions and functioning as a flexible hinge[43]. Ligand binding to this state likely triggers the conformational change in vivo[16]. The swinging motion is mediated by a hinge point formed between engaged proton relay residues, D407, D408, and K940 in AcrB (K938 in MdtF). This 'swing out' mechanism is important in enabling proton translocation from the cytoplasmic and periplasmic sides of the membrane, which powers substrate transport. Collectively, therefore, our structural analysis suggests that, at physiological pH at least, MdtF adopts similar porter domain transitions to AcrB during its functional rotation but that these transitions impart reduced, simultaneous movements within its

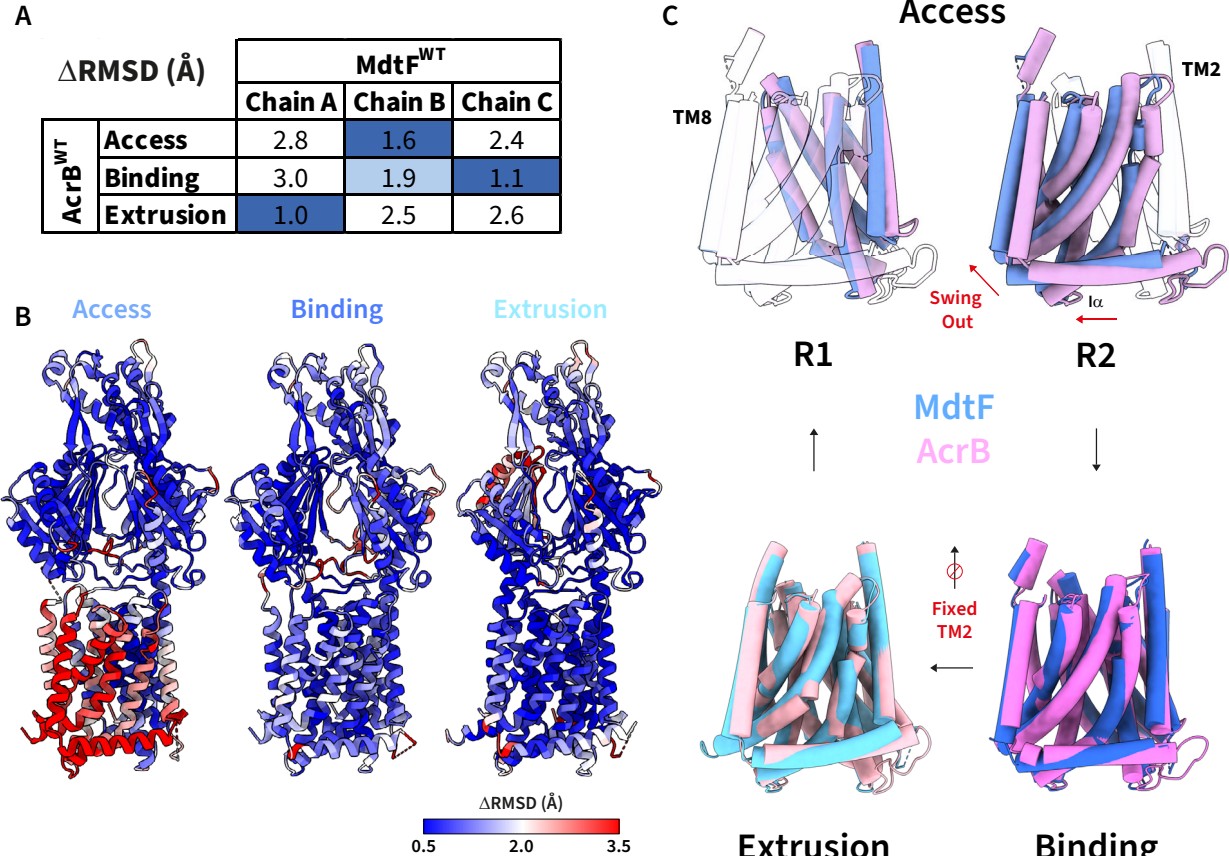

| ΔRMSD (Å) | | MdtF$^{WT}$ | | |
|---|---|---|---|---|
| | | Chain A | Chain B | Chain C |
| AcrB$^{WT}$ | Access | 2.8 | 1.6 | 2.4 |
| | Binding | 3.0 | 1.9 | 1.1 |
| | Extrusion | 1.0 | 2.5 | 2.6 |

**Fig. 2 | MdtF has a fixed 'swung-out' R2 transmembrane conformation across access and binding states. A** Each chain was assigned to the access, binding, or extrusion protomer state within AcrB (PDB: 4DX5)[38] according to global RMSD measurements (Cα-atoms). **B** Localised backbone RMSD calculations (Cα-atoms) reveal regional differences between AcrB and MdtF. **C** Alignment of AcrB (pink) and MdtF (blue) transmembrane domains, which reveal the structural differences between its helical arrangement as it cycles through the protomeric states. RMSD root mean square deviation, TM transmembrane helix.

transmembrane domain (the area responsible for the energy supply to these secondary transporters), especially for R2 region motion between access-to-binding states.

**Drug-binding domain plasticity directs MdtF drug-specificity**

Available, and allowable, drug channel and binding pocket conformations within RND efflux pumps act to define their polyspecific substrate efflux profiles[44]. Both residue patterning of substrate channels and entrances and conformational coupling have key roles in substrate selectivity across other homologues (Supplementary Discussion, Supplementary Tables 3 and 4 and Supplementary Figs. 14–18)[9,16]. To better understand the conformational determinants in MdtF, we solved the cryo-EM structure of a single-point mutation with an altered multidrug substrate profile (Val610Phe; MdtF$^{V610F}$) at a resolution of 3.28 Å, following the same procedure used for MdtF$^{WT}$. The Val610Phe mutation conferred increased resistance to linezolid and tetracycline but reduced tolerance to azithromycin, telithromycin, and the efflux pump inhibitor, phenylalanine-arginine β-naphthylamide (PAβN)[29]. This residue exists upstream of the switch loop and protrudes within the DBP of MdtF and exhibits a significance in substrate selectivity and transport (Fig. 3A). To confirm the altered MDR phenotype within our studies, we employed a minimum inhibitory concentration (MIC) assay on plasmid-borne MdtEF$^{V610F}$ in *E. coli* Δ9-Pore cells. These cells facilitate a direct assessment of substrate-related resistance in the absence of competing outer membrane diffusion. Here, we validated the relative alterations to phenotypic resistance arising from the single-point mutant (Fig. 3B) and ensured our purified protein is functionally congruent with previous studies[29].

The MdtF$^{V610F}$ structural model displays comparable porter domain asymmetry and altered access state transmembrane arrangements to MdtF$^{WT}$ (Supplementary Fig. 18). However, the V610F conversion engenders a deepened hydrophobic pocket within the DBP (Fig. 3C). To explore this affected pocket volume and better understand the mechanism of substrate binding and translocation, we aimed to solve a substrate-bound structure of MdtF$^{WT}$ and MdtF$^{V610F}$ with the known planar aromatic cation (PAC) substrate Rhodamine 6G (R6G). Using a fluorescence polarisation assay[45], we can exploit R6G fluorescence to monitor substrate binding. As free R6G rapidly rotates in solution, it will emit a weakened polarisation signal. Once an MdtF-R6G complex is formed, slower rotation of R6G will result in an increase in polarisation signal and, therefore, report on substrate binding. Here, we demonstrated that the purified protein is ligand recognition competent with similar binding affinity of the fluorescent R6G to MdtF$^{WT}$ and MdtF$^{V610F}$ ($k_D$ ~ 0.4–0.7 μM, Fig. 3D). Supporting thermal melting biophysical techniques, including circular dichroism (CD), and differential scanning calorimetry and fluorimetry (DSC and nanoDSF) showed that R6G binding had a similar stabilising effect on the periplasmic drug-binding domain within both constructs (Supplementary Figs. 20 and 21).

Although equilibrated with the same R6G concentration there was no observable, unambiguous density corresponding to R6G in the binding pocket of MdtF$^{WT}$, but we were able to solve the structure of a R6G-MdtF$^{V610F}$ complex at a resolution of 3.20 Å (Fig. 3E). R6G within our bound structure exclusively binds through hydrophobic interactions and is stabilised by cation-π (R6G-Phe626) and π-π (R6G-Phe178) interactions (Fig. 3E). As similarly observed in other substrate-bound

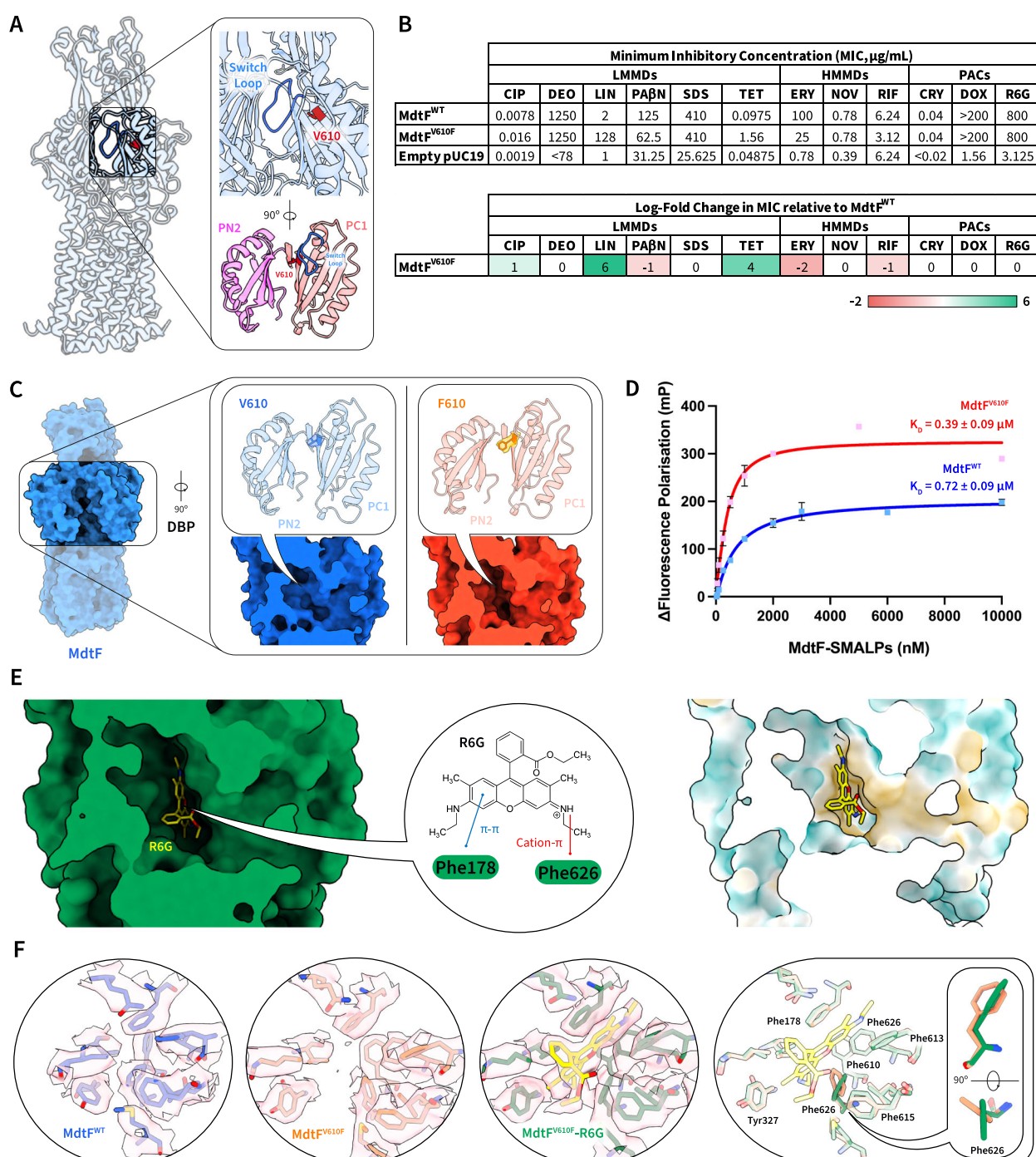

**Fig. 3 | V610F creates a hydrophobic nook within the DBP of MdtF. A** Location of the V610F residue observed within our cryo-EM structure for MdtF$^{WT}$. The switch loop and V610 residue is indicated in blue and red, respectively. The residue is found immediately upstream of the switch loop and protrudes into the DBP between the PC1 and PN2 subdomains. **B** MIC calculations to demonstrate the altered MDR phenotype because of the V610F mutation within MdtF. The values and log-fold change values are displayed to demonstrate the relative effects within *E. coli* Δ9-Pore cells. **C** Structure of apo-MdtF$^{V610F}$ solved by cryo-EM at a resolution of 3.28 Å with an observed formation of a 'hydrophobic nook' within the DBP, arising due to the single-point mutation. **D** Binding of R6G by MdtF$^{WT}$ and MdtF$^{V610F}$ as determined by a fluorescence polarisation assay. R6G was maintained at 1 μM throughout, and its emission wavelength was 550 nm. The binding isotherms demonstrate a $K_D$ of 0.39 ± 0.09 and 0.72 ± 0.09 μM for MdtF$^{WT}$ and MdtF$^{V610F}$, respectively, in a buffer containing 50 mM sodium phosphate, 150 mM NaCl, 10% glycerol. Reported data are the average and standard deviation from independent measurements ($n$ = 3) and were fitted to a hyperbola function (FP = (Bmax * [protein])/($K_D$ + [protein])). **E** Structure of R6G-MdtF solved by cryo-EM at a resolution of 3.20 Å, demonstrating cation-π (R6G-Phe626) and π-π (R6G-Phe178) interactions with R6G in the hydrophobic binding pocket of MdtF (colouring on the molecular surface from dark cyan (most hydrophilic) to white to dark golden (most lipophilic)). **F** MdtF$^{V610F}$ demonstrates a more open cleft within the binding pocket compared to MdtF$^{WT}$. R6G binding is facilitated by a reorientation of the Phe626 residue. DBP distal binding pocket, FP fluorescence polarisation, MIC minimum inhibitory concentration, R6G Rhodamine 6G.

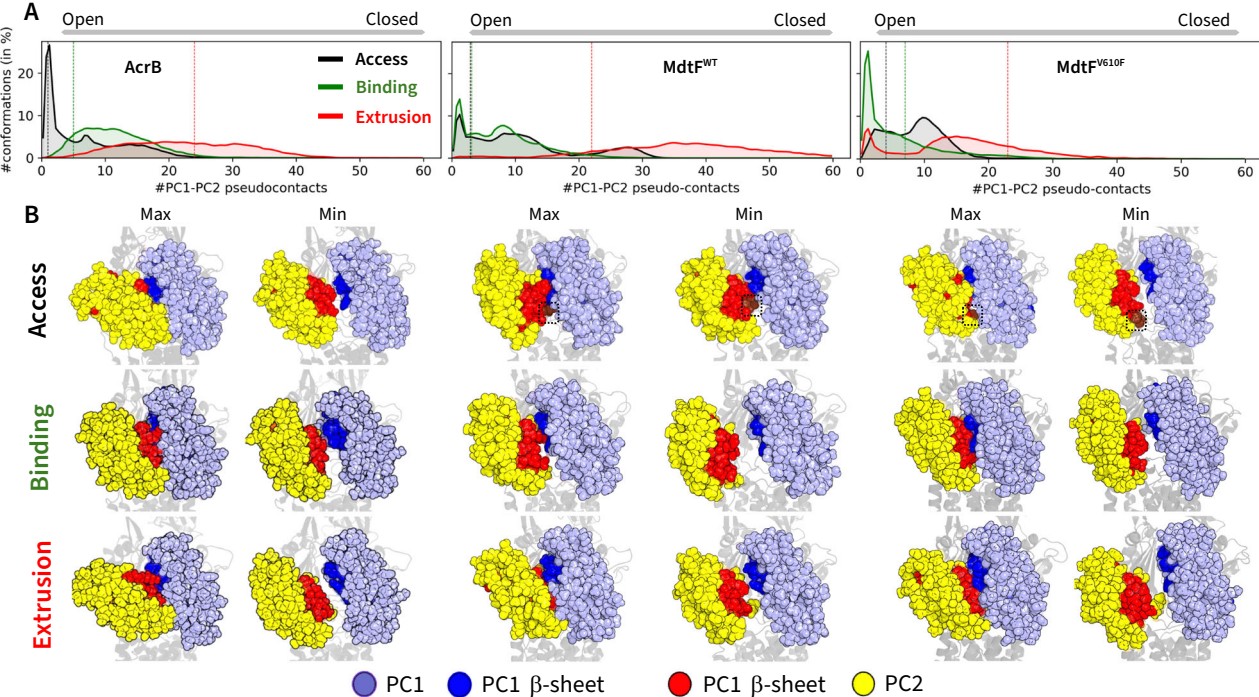

**Fig. 4 | Molecular dynamics reveals PC1-PC2 domain transformations are similar between AcrB and MdtF[V610F], but different in MdtF[WT]. A** Cumulative average of the pseudo contacts at the PC1-PC2 cleft, calculated using a cut-off of 10 Å between $C_a$ atoms and averaged from the simulation replicas of AcrB, MdtF[WT] and MdtF[V610F] (left to right). This plot represents the distribution of residue-residue pseudo contacts throughout the simulation trajectory. Contacts in the experimental reference structure are highlighted. Lesser contacts correspond to the cleft opening whereas higher number of contacts correspond to closing. **B** Extreme conformations of the PC1-PC2 cleft observed in the MD simulations. PC2 is represented as yellow spheres, PC1 is shown in light blue, and the beta sheets of PC1 and PC2 are depicted in blue and red, respectively. Note that MdtF[WT] and MdtF[V610F] has distinct orientation of A675 and S676 in the PC2 domain for 'access' protomer, as shown by the brown surface and highlighted by the black rectangle.

RND transporter structures R6G was found within the 'binding' promoter only[46], where a subtle rotation of the Phe626 ring accommodates binding within the DBP (Fig. 3F). Although possessing a similar overall affinity ($k_D$), the lower polarisation signal observed for R6G bound to MdtF[WT] could represent a binding mode that is less structurally restricted or heterogeneous, which could rationalise the absence of unambiguous ligand density in our dataset for R6G-MdtF[WT]. This notion is reinforced by the observation of the contracted pocket volume in MdtF[WT] which is penalised by steric clashes arising from the reorientation of phenylalanine rings F613, F626, F610, and Y327 which likely impede R6G ligand ordering within the pocket (Fig. 3F), thereby possibly preventing the elucidation of unambiguous R6G ligand density in MdtF[WT]. Collectively, these results suggest that the V610F mutation creates altered ligand recognition profiles through the emergence of a 'hydrophobic nook', possibly enabling MdtF to engage differently with its substrates.

Computational modelling was then performed to provide further insights into substrate interactions and conformational dynamics for the MdtF transporter. First, we performed molecular dynamics (MD) simulations for *apo* MdtF[WT] and MdtF[V610F] (as well as *apo* AcrB to compare dynamical features of the two transporters) in a model POPE:POPG (2:1) bilayer. Our investigations reveal no global level RMSD differences (of $C_\alpha$ atoms) across each 1 μs simulation (Supplementary Fig. 22); however, significant local level conformational transformations in both the porter and the transmembrane domains were detected. In the porter domain, the PC1 and PC2 subdomains of the entrance cleft in the access state of MdtF[WT], which define the entrances to channel 1 and 2 (CH1 and CH2) and mediate the uptake of several substrates in AcrB[44,47], displayed different preferred arrangements, as highlighted by the distribution of their pseudo-contacts (Fig. 4; see 'Methods' for details). Compared to AcrB, MdtF[WT]

has an overall more closed access state PC1-PC2 cleft at equilibrium (Fig. 4A). In particular, the distributions of pseudo-contacts almost overlap between the access and binding states of MdtF[WT] in contrast to AcrB, where the access state features a more open cleft (Supplementary Fig. 23), further supporting our classification of a 'binding-like' access state in MdtF. In addition, MdtF[WT] samples a greater number of distinct porter-domain conformations than AcrB, as indicated by the number of local maxima (peaks) in the pseudo-contact distributions−three vs. two peaks in the access state and two vs. one in the binding state (Fig. 4B)−reflecting a broader range of intermediate and extreme cleft conformations. We subsequently considered whether the creation of the 'hydrophobic nook' induced by the V610F mutation also leads to long-range conformational changes within MdtF. The cryo-EM structure of MdtF[V610F] reveals that the mutation partly alters the arrangement of PC1 and PC2 domains between the access and binding states, mainly arising from a shift in the distribution of the binding protomer. The extrusion state in MdtF[V610F] is characterised by a wider cleft compared to MdtF[WT] (Fig. 4A). Such a difference may partly be ascribed by the protrusion of residues A675 and S676 in the access state of MdtF[WT], seen in the cryo-EM structure of the MdtF[WT] transporter but not in MdtF[V610F]. Interestingly, MD simulations reveal that the V610F mutation induces an overall opening of the cleft in all monomers, resulting in a somewhat more symmetric MdtF conformation compared to the wild-type transporter. Notably, although our MD simulations reveal relatively large structural departures in the MdtF dynamics compared to the cryo-EM geometries for PC1-PC2 cleft (Fig. 4), the simulations of AcrB provided findings that are in good agreement with experimental data[16]. This supports the accuracy of our computational results and the possibility of a truly distinct mechanism of coupling between proton translocation and substrate export in MdtF.

Next, we performed molecular docking of known substrates linezolid, R6G, ciprofloxacin, and PAβN, as well as its natural nitrosyl indole substrate, on both MdtF$^{WT}$ and MdtF$^{V610F}$ to assess the impact of the mutation on possible direct interactions with substrates. We detected a semi-quantitative preference for all substrates to channel 3 (CH3) entry sites which was maintained in both MdtF$^{WT}$ and MdtF$^{V610F}$. Interestingly, there were very minor changes in the estimated affinities within the DBPs of MdtF$^{WT}$ and MdtF$^{V610F}$ (Supplementary Table 3). The largest change was the increase in affinity for ciprofloxacin within MdtF$^{V610F}$, which is compatible with enhanced efflux compared to MdtF$^{WT}$. Notably, the affinity of MdtF$^{WT}$ to R6G and the natural nitrosyl indole substrate remained unchanged in comparison to MdtF$^{V610F}$, suggesting their export would be unaffected in accordance with the fluorescence polarisation assay for R6G (Fig. 3D).

Together, our molecular dynamics and docking experiments suggest that MdtF adopts a smaller access state PC1-PC2 crevice opening than AcrB and a possible preference for CH3 mediated export (see Supplementary Discussion). CH3 directly leads to the DBP, bypassing the PBP and, in AcrB, it has been identified to prefer entry of planar aromatic cations (PACs), which could allude to the functional importance of MdtF within extreme acidic conditions where cationic substrates may become prominent. Further work exploring the structure-function relationships between MdtF and its known substrate classes will be required to fully explain how its conformations and residue patterning underpin specific substrate transport pathways. However, our data support the occurrence of allosteric conformational change resulting from a single-point V610F mutation and the importance of this residue in determining substrate specificity in related RND transporters[30].

## MdtF pump-identity is defined by its proton relay-related movements

The putative transmembrane proton relay network in MdtF (AcrB) comprises D407, D408, K938 (K940), and R969 (R971) (Fig. 5A). These residues are conserved across the RND transporters and are deemed essential for eliciting PMF-driven structural changes during efflux (Supplementary Fig. 24). To confirm the importance of the proton relay network in MdtF efflux, we employed a Nile Red efflux assay on a proton relay residue mutant D408A (MdtF$^{D408A}$). Here, *E. coli* Δ9-Pore cells lacking all nine TolC-dependent transporters were used to express plasmid-borne MdtEF. A decrease in fluorescence intensity is indicative of active efflux of the lipophilic dye Nile Red. In comparison to MdtEF$^{WT}$, MdtEF$^{V610F}$, and AcrAB$^{WT}$, MdtEF$^{D408A}$ eliminated efflux capability with no characteristic one-phase exponential decay observed (Fig. 5B), substantiating the importance of the putative proton relay network in MdtF function (Supplementary Fig. 14D). Intriguingly, MdtEF$^{WT}$ and MdtEF$^{V610F}$ ($t_{efflux50\%}$ = ~21 and ~23 s, respectively) rate of Nile Red efflux was observed to be slower in comparison to AcrAB$^{WT}$ ($t_{efflux50\%}$ = ~9 s).

We described earlier that the overall transitions and movements of transmembrane helices between the MdtF protomers are reduced compared to AcrB (Fig. 2C). We propose that this could, therefore, direct the mechanism by which MdtF can utilise, and transfer, the energy stored in the transmembrane electrochemical gradient to enact conformational changes for drug translocation throughout the porter domain. Our cryo-EM structures represent static interpretations and do not account for dynamic residue movements. By analysing the proton relay residue interactions occurring along MD trajectories, we observed a small depletion in the engagement of D407-K938 in the binding state of MdtF$^{WT}$ and MdtF$^{V610F}$ compared to AcrB (Fig. 5C, D). This is compensated by a non-vanishing engagement in the extrusion protomer, as seen in multiple independent simulations. The K938 residue is conserved across RND secondary transporters and has been postulated to be critical for proton sweeping through the proton-relay network across to the cytoplasmic membrane[48]. Furthermore, MdtF

features a relatively high occurrence of D407-R969 interactions in the access and binding states (Fig. 5D and Supplementary Figs. 25 and 26), which is absent in all AcrB protomers. It was proposed that the conserved R969 residue acts as a conformational electrostatic switch coupled to the protonation state of D407 and D408[16] to aid in efficient proton shuttling across the network. Together, these results point to enhanced engagement of D407 in the TM bundle of MdtF, which may hamper proton exchange through the proton-relay network and be the cause of the reduced efflux efficiency we observe versus AcrB, when in physiological pH conditions (Fig. 5B).

## MdtEF drug transport efficiency is responsive to pH

To better evaluate the effects of these transmembrane movements on MdtF-mediated efflux action, we performed accumulation assays with an increased range of substrates. We first determined whether the activity of MdtEF varies depending on the growth phase. When cells carrying an empty pUC19 vector or MdtEF$^{WT}$ were collected in the mid-exponential or stationary phases, the accumulation of Hoechst 33342 (HT) was lower in cells producing MdtEF$^{WT}$ than in cells with an empty vector. Thus, the pump is functional in both growing and stationary cells and can reduce the intracellular accumulation of HT.

Previous studies, however, showed that the outer membrane of *E. coli* and other bacteria is modified and becomes less permeable in the stationary non-growing cells[25]. Hence, such differences in the permeability barrier of the outer membrane could mask the differences in the efflux efficiency of MdtEF. Therefore, Δ9-Pore cells were induced with L-arabinose to produce a large non-specific Pore. Such hyperporination of the outer membrane eliminates its permeability barrier and enables a more specific assessment of efflux[49]. Surprisingly, we found that the exponentially growing and the stationary cells producing MdtEF differ in the accumulation of HT with the stationary cells accumulating significantly lower amount of HT than the exponential cells (Supplementary Fig. 27). In contrast, cells lacking the pump accumulated similar levels of HT in growing and stationary cells (Fig. 6A and Supplementary Fig. 27B). The levels of expression of MdtEF were similar in the compared cells (Supplementary Fig. 27A). However, the amounts of the Pore in the membrane fractions were notably lower in the stationary phase cells, suggesting that both the differences in the permeability of the outer membrane and differences in the activity of MdtF could contribute to the observed differences in HT accumulation in the stationary cells producing MdtF (Supplementary Fig. 27B).

We next analysed how the external pH affects the activity of MdtEF. Since HT is charged and its fluorescence is pH-sensitive (pK$_a$: N$_1$ 7.87, N$_2$ 5.81, and N$_3$ 5,02), in these experiments we used a non-ionisable membrane probe *N*-phenyl-naphthylamide (NPN). The exponential hyperporinated cells with and without MdtEF expression were resuspended in buffers with pH values decreasing from 8.0 to 5.0. We recently found that AcrAB retains similar NPN efflux activity across this identical pH range[50]. In contrast, the activity of MdtEF was highly pH-dependent and was strongly enhanced at acidic pH (Fig. 6B). Considering the environments which relate to the expression pattern of MdtF, these results signify the adaptation of its activity to become activated in an acidic periplasmic environment.

## Discussion

*Escherichia spp.* bacteria encounter hostile conditions during their enteric journey, such as oxygen and nutrient deprivation, as well as highly acidic environments, within the mammalian gut. The *E. coli* multidrug efflux pump MdtEF has been found to contribute to antimicrobial, biocide, and toxic dye resistance[42,51], as well as to bacterial fitness within anaerobic, acidic and macrophage environments[18,42,52]. MdtF is the multidrug transporter component of the pump, responsible for drug recognition and translocation, but its molecular mechanism is yet to be elucidated. It shares high sequence similarity

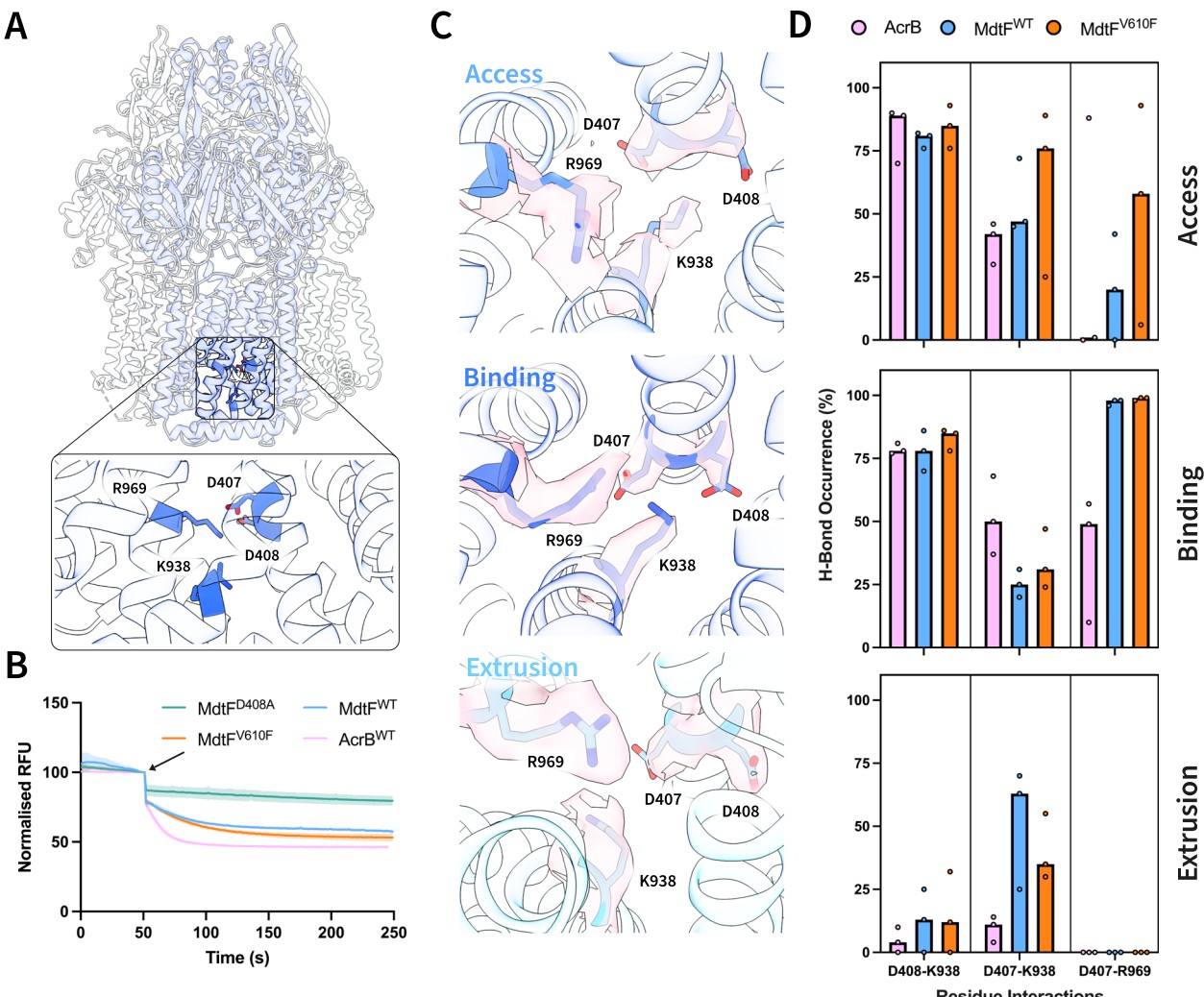

**Fig. 5 | Proton relay network residue D407 is hyper-engaged across protomer states of MdtF.** **A** Location of four essential proton relay residues (D407, D408, K938, and R969) within the transmembrane domain of RND transporters. **B** Nile Red efflux assay, performed in PBS buffer (pH 7.4), demonstrates the reduced rate of MdtEF-mediated efflux compared to AcrAB, and the MdtEF$^{D408A}$ mutant renders the transporter non-functional. *E. coli* Δ9-Pore cells harbouring p151A-AcrAB, pUC19-MdtEF$^{WT}$, pUC19-MdtEF$^{V610F}$, or pUC19-MdtEF$^{D408A}$ plasmids were monitored for fluorescence decay at 650 nm for 200 s. Here, proton gradient-powered efflux pumps were restarted following the addition of 1 mM glucose after 50 s (indicated by the arrow). Reported data are the average and standard deviation from independent measurements ($n = 3$) and were fitted to a one-phase decay exponential function (Y = (Y$_0$ − Plateau) * exp(−K * X) + Plateau). **C** Proton relay residues of

MdtF$^{WT}$ between each of the access, binding, and extrusion states depicted in a cartoon representation. The corresponding density for each residue is depicted in surface form (light pink). **D** Median H-bond occurrence (in % over the simulation time) at the proton relay site between access, binding, and extrusion states from top to bottom. The circles are indicative of an individual data point, which represents the average hydrogen bond occupancy over each of the three repeated simulations. The bar charts represent the median H-bond occurrence across three repeated simulations. H-bonds were detected using 3.0 Å between acceptor and donor, and an angle smaller than 35° between H, donor, and acceptor (calculated using the cptraj module of Amber24). For K938 and R969, the corresponding residues in MdtF are K940 and R971, respectively. *RFU* relative fluorescence units.

(~ 71%) and a phylogenetic clade with the constitutively expressed, prototypical RND transporter AcrB, as well as overlapping substrate polyspecificity, but is distinctly highly upregulated in anaerobic and early/late stationary phase conditions (~ 20- and 14/41-fold, respectively), raising questions on its functional implications.

Here, by combining cryo-EM structural information with functional assays and molecular docking and dynamics, we collectively establish a structural basis for MdtF-mediated substrate transport. We define a trapped transmembrane configuration between 'access' and 'binding' states in our structural model for MdtF, indicative of a unique RND functional rotation transport mechanism. Our molecular dynamics simulations identify a more closed PC1-PC2 cleft in the access state, which encompass the putative substrate channel 1 and 2 entrance sites, in comparison to AcrB. This, partnered with a suggested preference for substrates to interact with the channel 3 entrance

(located in the central cavity), hint at a conformational-induced preference for channel 3-utilising drugs. Interestingly, we reveal that a V610F mutation in MdtF (MdtF$^{V610F}$) altered its substrate specificity through drug-binding pocket expansion and PC1-PC2 domain engagement adjustments but retained similar transmembrane protomer state conformations and rate of efflux to MdtF$^{WT}$. This supports that when single-point mutational alterations to drug recognition occur, the mode of operation for powering their export is likely to remain consistent. MdtF is currently the only identified RND secondary transporter to elicit a defined physiological role in anaerobic conditions. Therefore, these results may serve as a mechanistic paradigm for other, related membrane protein transporters with anaerobic-associated expression and function.

Nevertheless, it remains unclear whether MdtF is important for expelling specific substrates that are produced under challenging

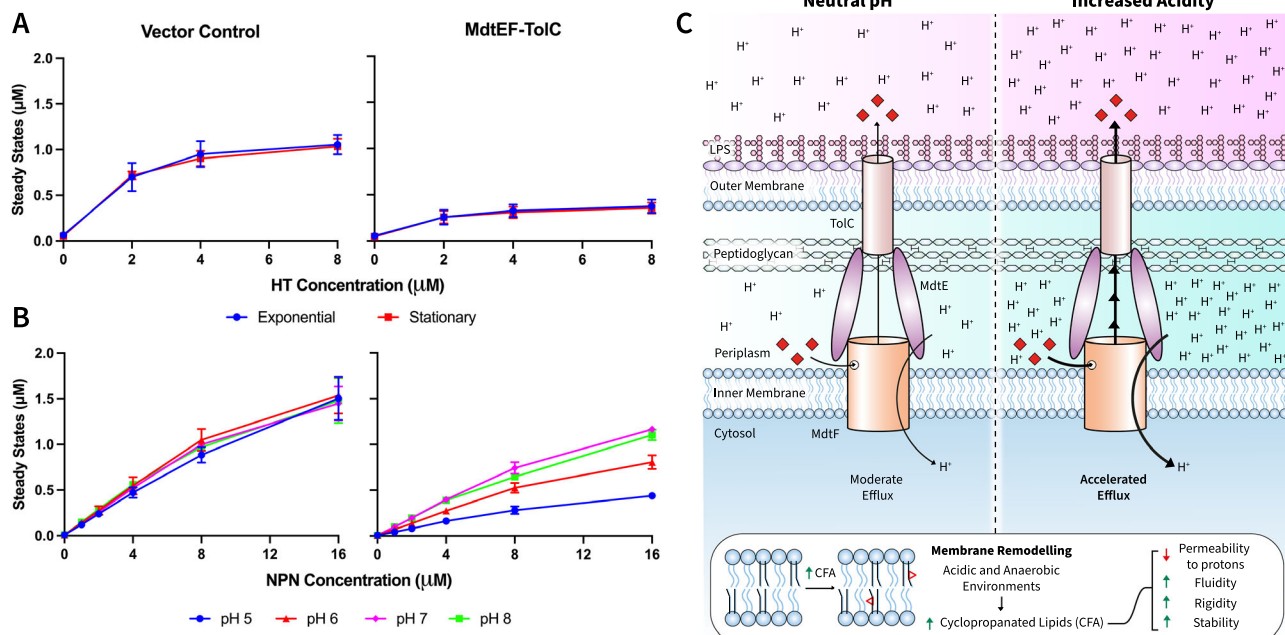

**Fig. 6 | MdtEF-TolC has acid-activated efflux.** Steady state accumulation levels of **A** HT and **B** NPN were measured in *E. coli* Δ9-pore cells, harbouring an empty vector or pUC19-MdtEF$^{WT}$ plasmid, as a function of external substrate concentration. Steady state accumulation levels were measured comparing either: **A** exponential or stationary phases or **B** differences in pH, ranging from pH 5 to 8. This was performed in the **A** absence or **B** presence of outer membrane Pore expression, due to differences in Pore expression at exponential and stationary phases (Supplementary Fig. 27A). Individual data points represent mean values from three independent measurements, and error bars are indicative of the standard deviation (*n* = 3). **C** Schematic of MdtEF-TolC acid-activated efflux in a CFA-remodelled inner membrane. MdtEF-TolC can provide bacterial defence and toxin disposal mechanisms for a range of bacterial environments, but seems to be a less efficient

system than the constitutively expressed AcrAB-TolC pump under physiological conditions. *mdtEF* is markedly, significantly upregulated in environmental settings that coincide with increased cyclopropanation of unsaturated inner membrane lipids, for example, under anaerobic stress and transition to stationary phase, which likely makes this its natural milieu. Both MdtF and CFA seem to have overlapped roles in maintaining bacterial fitness to acid shock. *mdtEF* becomes activated in response to acid to become much more efficient at substrate efflux. Where CFA lipids significantly change the membrane properties, leading to decreased permeability to protons, alongside increased membrane fluidity, rigidity, and stability. CFA cyclopropanated fatty acid, HT Hoechst 33342, NPN N-Phenyl-1-naphthylamine.

conditions, which need to be expelled, or if it is the conditions themselves that enable the function of MdtF. The acid-triggered activity discovered for *mdtEF* is intriguing. This acid-sensitive efflux behaviour has not yet been identified in other RND pumps and could rationalise the observed upregulation patterns of *mdtEF* in nutrient and oxygen starved bacterial populations, i.e., in the stationary phase and biofilms[23,24]. *mdtEF* is included on the Gad acid fitness island, sharing an operon with *gadE*. Induction of *gadE-mdtEF* is thought to be rigged under many different circumstances, such as stationary-phase and anaerobic transitions, that presage—instead of reacting to—an encounter with extreme acid, where *E. coli* cannot grow[53]. Here, we demonstrate this phenomenon could be happening at the molecular level of MdtF transport action (Fig. 6C). Its molecular architecture seems to create a polyspecific transporter whose efflux becomes more efficient at acidic pH when the membrane proton gradient increases. This could be an important adaptation when considering that all PMF-driven efflux pumps confer energetic costs due to gene expression and pump energy expense. Perhaps, favoured by more modest PMF-driven activity at physiological pH, elevated copies of *mdtEF* are permitted under nutrient starvation conditions, where envelope remodelling dominates xenobiotic protection through decreased permeability (e.g., during stationary-phase growth in the intestine) and a strong PMF is putatively harder to maintain[25,54]. Which, in turn, could create a barrage of acid-responsive RND transporters for innate waste management and activated xenobiotic clearance during extreme acid-exposure (such as gastric activity).

Notable limitations of our study are that MdtF structures were solved at neutral pH and extracted into SMALPs from stationary

phase membranes only. Although this has facilitated a direct comparison with AcrB, due its predominant structural investigation and understanding at physiological pH, it would be ideal to solve these structures in an acidic environment to better understand MdtF structure-function. However, the SMA nanodisc used in this study to maintain MdtF in a native lipid environment is susceptible to acidic pH, and membrane protein purification efficiency with alternative acid-stable co-polymer nanodiscs is considerably lower[55]. CFA-enriched inner membrane compositions are concurrent with bacterial cell envelope remodelling and *mdtEF* upregulation in anaerobic, stationary phases, biofilms and acidic stress. We confirm the presence of native CFA lipids within the lipid belt of our SMA-encapsulated MdtF, which supports that the conformations discovered by cryo-EM reflect a ground state within its natural milieu. Future work should aim to achieve structural models of MdtF, as well as AcrB, in a range of lipid compositions and pH environments to fully comprehend RND transport structure and function in remodelled *E. coli* envelopes and identify the key residues that contribute to acid-activated efflux in MdtF.

More generally, our work may contribute to the design and understanding of efflux pump inhibitors (EPIs) to restore and maintain antibiotic efficacy[56]. It has been established that MdtF can contribute to bacterial MDR and growth-cessation resilience, therefore, its structural elucidation would assist structure-guided drug discovery, for anti-biofilm agents for example. Relatedly, new EPIs have recently been discovered that target the proton-relay network within the transmembrane regions of AcrB[57]; therefore, the illumination that a closely related RND efflux pump can possess distinct

transmembrane conformers and proton network engagements could be important for avoiding (or encouraging) off-target drug design of these promising EPI-types.

## Methods

### Reagents and chemicals
Unless stated otherwise, all common reagents and chemicals were obtained from Sigma-Aldrich.

### Plasmids and mutagenesis
An overexpression plasmid containing the wildtype *mdtF* sequence with a C-terminal 6xHistidine tag (MdtF-6xHis) was constructed from a pET15b-MdtF-sfGFP-6xHis plasmid[58]. In brief, the *sfGFP* sequence was deleted by site-directed mutagenesis, according to the Q5® Site-Directed Mutagenesis Kit (New England Biolabs) using forward and reverse primers that flank the *sfGFP* sequence.

These plasmids, harbouring *mdtEF* and its native promoter, were constructed for bacterial susceptibility and Nile Red efflux assays. In brief, the pUC19 plasmid (Thermo Fisher Scientific) was linearised with PstI and BamHI restriction enzymes (New England Biolabs). The *mdtEF* gene, including the 180 base pairs upstream flanking sequence, was then amplified from *E. coli* genomic DNA (Zyagen) by PCR. The primers used incorporated a C-terminal 6xHis tag for detection by anti-His immunoblotting. The amplicon was subsequently cloned into the linearised pUC19 vector using NEBuilder® HiFi DNA Assembly kit (New England Biolabs). For MIC and Nile Red efflux assays, the 6 x Histidine tag was deleted by site-directed mutagenesis, according to the Q5® Site-Directed Mutagenesis Kit (New England Biolabs) using forward and reverse primers that flank the 6 x Histidine sequence.

For V610F mutagenesis, a single-point mutation was introduced in the pET15b-MdtF-6xHis plasmid according to the Q5® Site-Directed Mutagenesis Kit (New England Biolabs). Single-point mutations Q37T, P100A, D408A, and V610F were also introduced into the pUC19-MdtEF plasmids through site-directed mutagenesis, according to the Q5® Site-Directed Mutagenesis Kit (New England Biolabs).

The final plasmid sequence within the aforementioned plasmids were confirmed by Sanger sequencing (Eurofins Genomics). Primers used are delineated in Supplementary Table 5. Plasmid sequences are provided in Supplementary Data 1.

### Overexpression and purification of MdtF
The pET15b-MdtF-6xHis plasmid, containing the sequence for MdtF^WT or MdtF^V610F, was transformed into C43(DE3) Δ*acrAB E. coli* cells. 7 mL of an overnight LB culture was added to 1 L of pre-warmed LB culture containing 100 μg/mL ampicillin and grown at 37 °C until an $OD_{600}$ of 0.6–0.8 was reached. The culture was incubated at 4 °C for 30 min and then induced with 1 mM IPTG and grown for 16–18 h at 18 °C. The cells were subsequently harvested by centrifugation at 4200 × *g* and 4 °C for 30 min and washed with ice-cold phosphate buffer saline (PBS).

Cell pellets were immediately resuspended in Buffer A (50 mM sodium phosphate, 300 mM sodium chloride, pH 7.4), supplemented with 1 μL of Benzonase, 5 mM beta-mercaptoethanol (β-ME), 1 mM phenylmethylsulphonyl fluoride (PMSF), and a protease inhibitor tablet (Roche). The cell suspension was then passed twice through a microfluidizer processor at 25,000 psi and 4 °C. Insoluble material was removed by centrifugation at 20,000 × *g* and 4 °C for 30 min. Membranes were pelleted from the supernatant by centrifugation at 200,000 × *g* and 4 °C for 1 h. Membrane pellets were resuspended in 30 mL of ice-cold buffer B (50 mM sodium phosphate, 150 mM sodium chloride, 10% (w/v) glycerol, pH 7.4), supplemented with 1 mM PMSF and a protease inhibitor tablet (Roche), and homogenised using a Potter-Elvehjem Teflon pestle and glass tube.

MdtF was extracted from homogenised membranes by the addition of 2.5% (w/v) SMA 2000 co-polymer powder (Cray Valley) and incubation for 2 h at room temperature with gentle agitation.

Insoluble material was then removed by centrifugation at 100,000 × *g* for 1 h at 4 °C. 1 mL Ni-NTA resin (Generon or Thermo Fisher Scientific), equilibrated in Buffer B with 20 mM imidazole, was added directly to the supernatant and incubated overnight at 4 °C with gentle agitation. The sample was then transferred to a gravity flow column (Bio-Rad) and washed with 20 column volumes of Buffer B with 20 mM imidazole and 10 column volumes of Buffer B with 50 mM imidazole. MdtF was eluted with 5 column volumes of Buffer B with 500 mM imidazole. The eluted protein was filtered through a 0.22 μm filter membrane (Thermo Fisher Scientific) and loaded onto a Superdex 200 Increase 10/300 GL SEC column (GE Healthcare) equilibrated in Buffer B. A flow rate of 0.4 mL/min was used during SEC purification. Peak fractions eluted from the SEC column containing pure MdtF were pooled, spin concentrated using a 100 K MWCO Vivaspin® 6 spin concentrator (Sartorius), flash-frozen in liquid nitrogen and stored at − 80 °C. Samples were separated by SDS-PAGE in a 12% NuPage™ Bis-Tris Precast Gel (Thermo Fisher Scientific) to assess MdtF purification. Gels were run at 180 V and room temperature for 1 h. Protein concentration was calculated using a NanoPhotometer™ N60 UV/Vis spectrophotometer (Implen) with an extinction coefficient of $\varepsilon_{280} = 86{,}305\,M^{-1}cm^{-1}$.

### DLS measurements
10 μL of the sample was diluted in 990 μL of deionised water in a 4.5 mL disposable cuvette. Dynamic light scattering (DLS) measurements were performed on a Litesizer™ 500 (Anton Paar) at 25 °C for 20 processed runs. The size distribution in radius is displayed on a logarithmic scale.

### Negative stain EM analysis
MdtF-SMALPs were initially filtered through a 0.22 μm Spin-X centrifuge tube filter (Costar) and buffer exchanged into a Tris-based Protein Buffer (50 mM Tris, 150 mM sodium chloride, pH 7.4) using Micro Bio-Spin 6 columns (Bio-Rad). Samples were then spin concentrated using a 100 K MWCO Vivaspin® 500 spin concentrator (Sartorius).

Negative stain grids were prepared by applying 3 μL of the respective protein samples onto a previously glow-discharged carbon-coated copper (300 mesh) grid. The grids were then stained with 3% uranyl acetate for 1 min. The grids were air dried for 2 min and imaged using a JEM-1400 FLASH (JEOL) TEM equipped with a tungsten filament operating at 120 kV. Digital images were acquired at a nominal magnification of ×60,000 with a bottom-mounted FLASH 2 k × 2 k CMOS camera.

### Cryo-EM sample preparation and data acquisition
MdtF samples were filtered through a 0.22 μm Spin-X centrifuge tube filter (Costar) and buffer exchanged into a Tris-based Protein Buffer (50 mM Tris, 150 mM sodium chloride, pH 7.4) using Micro Bio-Spin 6 columns (Bio-Rad). For the R6G-bound state, MdtF^V610F samples were treated similarly except for the addition of 75 μM R6G followed by a 4 h incubation on ice. To prepare grids for vitrification, 4 μL of apo-MdtF^WT at 0.25 mg mL⁻¹, apo-MdtF^V610F at 0.5 mg mL⁻¹, and R6G-MdtF^V610F at 0.5 mg mL⁻¹ were applied to previously glow-discharged UltrAufoil 1.2/1.3 gold or Quantifoil R1.2/1.3 holey carbon EM grids (Quantifoil Micro Tools). Cryo-EM samples were subsequently blotted using a Mark IV Vitrobot (Thermo Fisher Scientific) for 4 s with a blot force of 7 at 4 °C and 100% humidity prior to immediate plunge-freezing in liquid nitrogen-cooled ethane. Cryo-EM specimens were screened using a Glacios™ Cryo-TEM (Thermo Fisher Scientific) to examine particle density and ice quality.

For apo-MdtF^WT, 5675 movies were collected on a Glacios™ Cryo-TEM with a Falcon III detector, operating at 200 kV accelerating voltage and a nominal magnification of ×190,000, corresponding to a physical pixel size of 0.946 Å pixel⁻¹. The apo-MdtF^V610F and R6G-

MdtF$^{V610F}$ datasets were collected using a similar approach, except 8000 movies were recorded for both on a Falcon 4 camera with a physical pixel size of 0.947 Å pixel$^{-1}$ and a nominal magnification of ×150,000. Cryo-EM data collection for each dataset is summarised in Supplementary Table 1.

## Image processing

The image processing steps for each dataset are depicted in Supplementary Figs. 2, 4, and 6. In brief, movie stacks were pre-processed with RELION's implementation of MotionCor2[59,60] using a 5 × 5 grid (25 tiles), for motion correction and dose-weighting according to the implemented pre-calibrated filtering scheme. The micrograph-based contrast transfer function (CTF) was determined with CTFFIND4[61] on non-dose weighted, drift-corrected micrographs. A subset of micrographs with a CTF fit accurate up to 8 Å or better were selected. Particles were subsequently auto-picked in RELION-4.0[60] using Laplacian-of-Gaussian with a diameter between 100 and 150 nm. Particles were extracted with a box size of 260 × 260 pixels, after binning by a factor of two. Low quality particles and false-positive picks in the automatically picked dataset were identified by a reference-free 2D classification, separating particles into 200 classes. Of these 2D classes, representative classes were subsequently used for a reference-based particle picking in RELION-4.0[60]. Subsequent classifications and refinements were performed in RELION-4.0. For 3D classification, a map of AcrB (EMD-24653 [https://www.ebi.ac.uk/emdb/EMD-24653]) was low-pass filtered to 60 Å and used as an initial reference. 3D classification was performed with no symmetry imposed (C1). From three 3D classes, the particle images comprising the 3D class with better features corresponding to MdtF were subsequently un-binned to the original pixel size. To prevent the loss of any asymmetric structural features of MdtF, all steps were undertaken with no applied symmetry. Following Bayesian polishing, CTF refinement, and beam tilt estimation with RELION-4.0[60], refinement of this subset of particles with a soft mask and no applied symmetry (C1) produced maps at nominal resolutions of 3.56, 3.28, and 3.20 Å (gold-standard Fourier shell correlation (FSC) = 0.143) for the apo-MdtF$^{WT}$, apo-MdtF$^{V610F}$, and R6G-MdtF$^{V610F}$, respectively.

## Model building and structure refinement

A homology model was obtained for MdtF$^{WT}$ using SwissModel[62], which was used as a starting point for modelling with Coot (version 0.9.8.3)[63]. This model was subsequently used as a starting point for generating the atomic models for apo-MdtF$^{V610F}$ and R6G-MdtF$^{V610F}$. Models for lipids within the transmembrane region were modelled using Coot (version 0.9.8.3)[63] as previously described[31]. Briefly, PE or dodecane molecules were modelled into the structures with varying chain lengths, which corresponded to the electron density they occupied. Atomic models were refined in Phenix (version 1.20)[64] using real-space refinement and validated with Molprobity[65]. ChimeraX[66] was used for the visualisation of Cryo-EM maps and preparation of figures.

## Fluorescence polarisation assay

MdtF ligand binding was determined using fluorescence polarisation (FP) assays as performed by Su et al.[45]. MdtF protein titration experiments were conducted in ligand binding solution (50 mM sodium phosphate, 150 mM sodium chloride, 10% (w/v) glycerol, 1 μM R6G, pH 7.4). FP measurements were recorded following incubation for 10 min for each corresponding protein concentration to ensure binding had reached equilibrium. All measurements were performed at 25 °C and the excitation wavelengths were set to 525 nm and fluorescence polarisation signals were measured at an emission wavelength of 550 nm. Each data point was an average of 15 FP measurements and ligand binding data were fitted to a hyperbola function: $FP = \frac{B_{max} \times [Protein]}{K_D + [Protein]}$ using SigmaPlot (Version 15.0) as conducted by Su et al.[45].

## Nile Red efflux assay

E. coli Δ9-Pore cells[49] were transformed with either pUC19-MdtEF$^{WT}$, pUC19-MdtEF$^{V610F}$, pUC19-MdtEF$^{D408A}$, or p151A-AcrAB plasmids. An overnight LB culture was added to 50 mL of pre-warmed LB culture containing 100 μg/mL ampicillin at a starting OD$_{600}$ of 0.05. Cells were sub-cultured with 0.1% L-arabinose and grown at 37 °C and 5000 × g until an OD$_{600}$ of 1.0 was reached. Bacteria were then centrifuged and resuspended in 10 mL of PBS (pH 7.4), containing 1 mM MgCl$_2$ (PPB). Cells were pelleted again and resuspended in 10 mL of PPB. Samples were separated into 2 mL aliquots with an OD$_{600}$ of ~1.0 within 10 mL Pyrex® disposable glass conical centrifuge tubes to avoid Nile Red adherence to the walls. 10 μM of Carbonyl cyanide m-chlorophenylhydrazone (CCCP, Cambridge Bioscience) was added and were incubated for 20 min at 37 °C and 5000 × g. Following this incubation, 5 μM of Nile Red was added to cells and were incubated for an additional 1 h 30 min at 37 °C and 5000 × g. Cells were pelleted and washed twice with PPB buffer and further diluted 10-fold. Nile Red fluorescence was measured at 650 nm over 250 s, at 1 s intervals ($\lambda_{ex}$ = 544 nm, slit width of 5 nm; $\lambda_{em}$ = 650 nm, slit width of 10 nm) using a Fluoromax®-4 with FluorEssence™ (Horiba). 50 mM glucose was added after 50 s to re-energise the proton-powered pumps. The resulting fluorescence curves were normalised to a starting value of 1.0. All curves are an average of at least three independent runs undertaken with E. coli Δ9-Pore cells, which have been separately transformed with either pUC19-MdtEF$^{WT}$, pUC19-MdtEF$^{V610F}$, pUC19-MdtEF$^{D408A}$, or p151A-AcrAB plasmids.

## Minimal inhibitory concentration assay

Susceptibilities of the E. coli Δ9-Pore cells harbouring various plasmid-borne MdtF variants against ciprofloxacin, crystal violet, deoxycholate, doxorubicin (Fluorochem Limited), erythromycin, linezolid, novobiocin, PAβN, rhodamine 6 G, rifampicin, SDS, and tetracyline (Abcam) were determined by two-fold broth microdilution as previously described[67]. Briefly, overnight cultures were sub-cultured in LB broth (tryptone, 10 g/l; yeast extract, 5 g/l; NaCl, 5 g/l), and cells were grown at 37 °C in a shaker at 220 rpm until OD$_{600}$ reach to 0.2–0.3. For the proper expression of the pore, L-arabinose (final concentration of 0.1%) was added to each culture, and cells were further grown until OD$_{600}$ reached 1.0. The minimum inhibitory concentration of each bacterial strain against different antibiotics was measured in 96-well plates. Exponentially growing cells were added to each well and incubated for 18 h. European Committee on Antimicrobial Susceptibility Testing (EUCAST) guidelines were followed conforming to ISO 20776-1:2006[68,69]. Substrates were prepared and used according to the manufacturer's instructions. E. coli Δ9-Pore cells harbouring an empty pUC19 plasmid was used as the negative control.

## MdtEF expression test

The presence of MdtF and its single-point mutants for the MIC and Nile Red efflux assays were confirmed by Western blotting. In brief, an overnight culture of E. coli Δ9-Pore cells harbouring pUC19-MdtEF, or empty pUC19 plasmids were sub-cultured with 0.1% L-arabinose until they reached either exponential (OD$_{600}$ = 0.8) or stationary (OD$_{600}$ = 2.0) phase. Here, 1 mL of each culture was pelleted by centrifugation at 16,000 × g at 4 °C for 5 min and washed twice with PBS. The pellets were resuspended in 100 mL of ice-cold RIPA lysis buffer (1 mM EDTA, 1% Triton X, 0.1% sodium deoxycholate, 0.1% SDS, 140 mM NaCl, 10 mM Tris-HCl, pH 8.0), supplemented with 100 mM PMSF and a protease inhibitor tablet, and incubated at 4 °C with shaking for 10 min. The cells were subsequently sonicated for 15 min, and insoluble contents were removed by centrifugation at 16,000 × g at 4 °C for 5 min. The supernatant was aliquoted and mixed with SDS loading buffer prior to loading 20 μL on a 4–12% NuPAGE™ Bis-Tris Protein Gel (Thermo Fisher Scientific). For the detection of protein using western blotting, the protein was transferred to a nitrocellulose

membrane and an anti-His antibody was used to detect the His-tag containing proteins.

## SDS-PAGE

Proteins were diluted in 5 x Laemmli sample buffer (312.5 mM Tris, 50% glycerol, 100 mg/mL SDS, 80 mg/mL dithiothreitol (DTT), 0.1% bromophenol blue, pH 6.8). Samples were run on either a 4–10, 10, or 12% pre-cast NuPAGE™ Bis-Tris or Novex Tris-Glycine Gel (Thermo Fisher Scientific). Bis-Tris gels were run with a 20 x NuPAGE™ MOPS SDS running buffer (50 mM MOPS, 50 mM Tris Base, 0.1% SDS, 1 mM EDTA, pH 7.7). Tris-Glycine gels were run with a 10 x Tris-Glycine running buffer (25 mM Tris Base, 192 mM glycine, pH 8.3). The Novex Sharp Pre-Stained Protein Standard (Thermo Fisher Scientific) was used as a protein ladder for molecular weight estimation. Samples were run at room temperature for 1 h at 180 V, and protein bands were subsequently visualised using Coomassie® Brilliant Blue G-250 stain (VWR).

## Western blotting

For anti-His Western blotting, SDS-PAGE gels were transferred to a nitrocellulose membrane (0.2 µm pore size) membrane. Membranes were subsequently equilibrated in transfer buffer (25 mM Tris, 192 mM glycine, 0.77% (w/v) SDS, 10% (v/v) methanol) and proteins were then transferred from the cell to the membrane using a Cytvia Amersham™ TE 77 PWR semi-dry transfer unit (Thermo Fisher Scientific). Transfer was run at room temperature for 1 h at 45 A per gel. Following this, membranes were subsequently blocked in 5% (w/v) milk powder in PBS-Tween (PBS-t, 0.1% (v/v) Tween) for 1 h at room temperature or 4 °C overnight with gentle shaking. Membranes were further incubated with anti-polyhistidine-HRP antibody (1:10,000) dilutions for 1 h at room temperature or 4 °C overnight with gentle shaking. Membranes were subsequently washed with PBS-t four times at 15 min intervals. Membranes were then developed with 1 mL Amersham™ ECL Select™ solution for 1 min and imaged using an A1600 Imager (GE Healthcare).

## Native SMA-PAGE

Native SMA-PAGE was performed as previously described[70]. In brief, samples were run on a pre-cast Novex™ Value™ Tris-Glycine protein gel, 4–20% (Thermo Fisher Scientific). Protein samples were diluted in 2 x Novex™ Tris-Glycine Native sample buffer (Thermo Fisher Scientific) or 4 x NativePAGE sample buffer (Thermo Fisher Scientific). The NativeMark™ Unstained Protein Standard (Thermo Fisher Scientific) was used as a protein ladder for molecular weight estimation. 30 µL of protein was loaded per well and samples were run for 90 min at 150 V and 4 °C. Protein bands were subsequently visualised using Coomassie® Brilliant Blue G-250 stain (VWR).

## Lipid extraction

For LC-MS and GC-MS experiments, lipids were extracted according to a modified version of Bligh and Dyer[71]. In brief, lipid-containing samples (0.5 mL) were added to 1.7 mL chloroform: methanol: 1 M Tris at pH 8 (10:23:1 (vol/vol/vol)) and mixed extensively. To achieve phase separation, 1 mL of a 1:1 mixture of chloroform and 0.1 M Tris at pH 8 was added. The lipid-containing organic phase was then collected and evaporated under a stream of nitrogen to provide a total lipid extract film.

## GC-MS of extracted lipids from MdtF SMALPs

GC-MS was performed as previously described[72]. Fatty acyl methyl esters (FAMEs) were prepared by derivatisation of the lipid-containing samples for identification of CFAs using GC-MS. In brief, 0.5 mg of dry lipid extract was dissolved in 100 µL toluene, 750 µL methanol, and 150 µL 8% HCl solution. Following an hour incubation at 100 °C, 0.5 mL hexane and 0.5 mL water was added. The mixture was then vortexed and centrifuged at 6000 × g for 5 min. The FAME-containing organic phase was separated, and the aqueous phase was re-extracted with 250 µL of hexane. The organic phases were combined and used for fatty acid analysis by GC-MS.

GC-MS analysis was conducted with a Shimadzu Nexus GC-2030 system equipped with a Shimadzu QP2020 NX GC-MS single quadrupole spectrometer, with a Shimadzu AOC-6000 autosampler. The FAME mixture was separated using an SH-RXI-5MS column (30 m × 0.25 mm × 0.25 µm, Shimadzu). The temperature gradient was as follows: 150 °C (4 min); 4 °C/min to 250 °C (11 min). The carrier gas was helium with a linear flow rate of 25.5 cm/s. The injector temperature was 250 °C and the injection volume was 1 µL with a 10:1 split. Detection was performed using Selected Ion Monitoring (SIM), and a commercial mixture of bacterial acid methyl esters was used for identification of fatty acids based on their retention time (RT).

The raw data were processed using GCMSsolution software (version 4.52, Shimadzu Scientific Instruments, Kyoto, Japan). Signal peaks were identified using ions $m/z = 74$ collected in the SIM mode, and their RT and sample peak areas were calculated. Detected FAMEs within the SMA-extracted MdtF samples were identified according to those present and identified within a qualitative standard Bacterial Acid Methyl Ester (BAME) Mix (part number (47080-U) and Lot number (LRAD6478), Sigma-Aldrich) with a corresponding RT. All peaks identified within SMA-extracted MdtF samples were also observed within the bacterial fatty acid methyl ester samples. GC-MS values are presented as relative proportions within each sample.

## LC-MS analysis of extracted lipids from MdtF SMALPs

Lipids extracts were prepared with an internal standard mix as previously described[33]. All samples were analysed via liquid chromatography-mass spectrometry (LC-MS) using a Vanquish Flex LC (Thermo Fisher, Hemel Hempstead, UK) with an Acquity BEH C18 column (Waters, Wilmslow, UK) attached, coupled to a Q Exactive Plus mass spectrometer (Thermo Fisher, Hemel Hempstead, UK). Mobile phase A was composed of acetonitrile: water (60:40) with 10 mM ammonium acetate. Mobile phase B was composed of isopropanol: acetonitrile (90:10) with 10 mM ammonium acetate. An 18-min LC gradient was run at 0.3 µL/min, starting at 40% mobile phase B, increasing to 99% B at 10 min, and then decreasing back to 40% B at 13.5 min to the end of the run. The source was operated at 3.2 kV, capillary temperature 375 °C, and desolvation temperature and gas flow 400 °C and 60 L/h, respectively. All mass spectra were acquired in negative ion mode in the range of 150–2000 $m/z$. Data processing and analysis was performed using Expressionist software (version 15, Genedata, Basel, Switzerland), for chromatographic and spectral peak detection and isotope clustering. Identification of lipids was performed via accurate mass against the LIPIDMAPs database. The 250 most intense peaks within the spectrum were extracted to form a peak list, following chromatographic and spectral peak detection, charge, and isotope clustering. The extracted peak list was searched in LIPIDMAPs[73] using an error tolerance of ± 0.01 Da, against the glycerophospholipids class, with ion adducts $[M-H]^-$, $[M+OAc]^-$, $[M-CH_3]^-$ and $[M-2H]^{2-}$ selected. Lipid matches that were not biologically relevant were excluded from the search results, such as the exclusion of phosphatidylcholine (PC) as *E. coli* doesn't naturally produce PC phospholipids. The search was performed against the glycerophospholipid class. For confirmation of the identities of lipids, MS/MS was performed using HCD. A DDA method with collision energies of 10 eV, 20 eV and 30 eV were used for fragmentation of parent ions, and the data was combined for database searching. In Expressionist was used for library MS/MS searching against the LipidBlast database[74].

## Functional assays and immunoblotting normalisation

The HT and NPN uptake assays were conducted using *E. coli* Δ9-Pore cells complemented with either the pUC19 vector or its derivative carrying the gene encoding MdtEF^WT. The HT uptake experiment was performed with both exponential-phase cells, collected 4 h after

subculture, and stationary-phase cells, collected 20 h post-subculture. Overnight cultures of *E. coli* Δ9-Pore strains (pUC19 and MdtEF[WT]) were diluted 1:100 in fresh LB and incubated at 37 °C with shaking until reaching mid-log phase (OD$_{600}$ ~ 0.3–0.4). For conditions requiring pore induction, 0.1% L-arabinose was added, and the cultures were allowed to grow for either 4 h (exponential phase) or 20 h (stationary phase). In the absence of L-arabinose, cultures were grown for the respective time periods without induction. After incubation, cultures were adjusted to OD$_{600}$ ~ 1.0, centrifuged, and resuspended in HMG buffer (50 mM HEPES-KOH, 0.4% (w/v) glucose, 1 mM MgSO$_4$, pH 7.0) at the appropriate pH. The cells were then readjusted to OD$_{600}$ ~ 1.0 before proceeding with the uptake assay. For studies of the pH effect, induced exponential *E. coli* Δ9-Pore cells were collected by centrifugation, washed, and resuspended in the buffer adjusted to different pH values to final OD$_{600}$ ~ 1.0.

For the uptake assay, a 96-well black clear-bottom plate was prepared with increasing concentrations of HT as previously described[75]. Briefly, bacterial suspensions at OD$_{600}$ ~ 1.0 were added to each well (100 µl per well), and fluorescence measurements were taken immediately upon cell addition using a spectrophotometer. The fluorescence settings were optimised for HT (excitation: 355 nm, emission: 450 nm). Steady-state concentrations were calculated using MATLAB, and raw fluorescence data were analysed and plotted in Microsoft Excel and GraphPad Prism 10 for visualisation and comparison.

For protein expression analysis, aliquots of bacterial cell suspensions prepared for the uptake experiments above were harvested by centrifugation. Cells were lysed via sonication, and lysates were subjected to differential centrifugation, first at 4000 × *g* to remove debris, followed by ultracentrifugation at 40,000 × *g* to obtain membrane fractions. The total protein concentration was measured using the Bradford assay. Equal amounts of protein were resolved via SDS-PAGE and transferred onto a PVDF membrane for western blot analysis. The membrane was blocked with 1% BSA and probed with an anti-His primary antibody, followed by an anti-mouse alkaline phosphatase-conjugated secondary antibody.

### Differential scanning fluorimetry (DSF)

NanoDSF experiments were performed as follows. 10 µL of 2 mg/mL MdtF-SMALPs ± 75 µM R6G was loaded into standard capillaries and onto the capillary tray of a Prometheus Panta instrument (Nanotemper Technologies). The experiment was run at a temperature gradient from 10 to 100 °C at 1 °C/min, an excitation power of 7% and DLS power of 100%. The data for each channel was subsequently analysed using the PR.Panta Analysis Software (Nanotemper Technologies) to analyse thermal stability and particle size.

### Molecular docking

We investigated the binding of R6G, PAβN, linezolid, ciprofloxacin, and nitrosyl oxide at physiological pH. These studies were conducted for both the MdtF[WT] and MdtF[V610F] transporter (utilising two ensemble structures for each variant), focusing on the binding protomer using the cryo-EM structures of apo and holo states of MdtF[WT] and MdtF[V610F] (WT bound structure was generated using homology modelling by Prime using mutant V610F bound) for the distal pocket, whereas only apo structure were used for the channels docking. The protein structures of MdtF (obtained in this work) were prepared using the *Protein Preparation Wizard* in Maestro (Schrödinger Release 2024-1) with the default settings. This preparation involved adjusting for missing hydrogen atoms, incomplete side chains and loops, assigning appropriate protonation states to ionisable groups in proteins based on pKa values calculated with PROPKA3[76,77], and flipping relevant residues to reflect the physiological conditions[78]. Following this, we performed a restrained minimisation that permitted free movement of hydrogen atoms (threshold of <0.30 Å) while allowing sufficient relaxation of

heavy atoms to alleviate strained bonds and angles. The missing loops (residues 496–515) were modelled and refined using Prime based on reference UniProt sequence P37637 utilising OPLS4 force field and implicit solvation model VSGB[79–81]. In particular, at the proton relay site in transmembrane domain, protonation states were assigned following Eicher et al.[16], i.e., E346 and D924 were protonated only in access and binding protomers, while D407, D408, and D566 were protonated only in the extrusion protomer[16,82,83]. For the ligands modelling, possible ionisation states were generated using Epik (*Ligprep*) at pH 7.4[84]. Ligand conformation sampling for each pH condition was executed using Macromodel[85] (Mixed Mode). The optimised conformers of the five ligands were docked against the grid generated centring around the native ligand (Rhodamine 6G) for the distal pocket, and centred around specific residues for CH1 (T559, S834, G836, A838, K840, E864, L866, S868, Q870, P872), CH2 (E564, T643, Q647, I651, V660, T674, S676, G713, R715, N717, E720, L826, E828) and CH3 (A33, Q37, P100, G296, N298)[86] residues (see Supplementary Fig. 9B for a comparison of the residues lining these channels and the DBP in MdtF and AcrB) using standard precision (SP) mode[87,88]. Top 20 poses were further processed for the second docking run using extra precision (XP) mode[88]. Best poses corresponding to the Glide score were ranked and chosen to analyse the interaction details. Despite their widespread applications, docking methods such as Glide are known to have inherent limitations in accurately predicting absolute binding affinities, and are best interpreted for relative ranking and qualitative comparisons across ligands and receptor variants[89,90]. Prior to docking, the redocking (as a control) was performed to confirm the performance and accuracy of the protocol for the R6G in the DP. In molecular docking, for our entire ligand and receptor modelling, we adhered to OPLS4 force field[91]. The images were plotted using pymol[92].

### Molecular dynamics (MD)

MD simulations of apo proteins were performed for three systems: MdtF (MdtF[WT] and MdtF[V610F]), and the well-established AcrB structure in its asymmetrical state (PDB ID: 4DX5)[38]. The structure of AcrB was prepared following the same protocol described for MdtF in the docking section. Three chains of consistent lengths used and corresponding residues for AcrB/MdtF were protonated as per PROPKA3 in all three states at physiological pH[82,83]. For MdtF (AcrB) residues E346 (E346) and D922 (D924) were protonated only in the access and binding protomers, while residues D407 (D407), D408 (D408), and D566 (D568) were protonated only in the extrusion protomer. Amber24 was used for the system setup and simulation[93,94]. The structures were converted to GAFF2 atom types using pdb4amber (Ambertools23), during which hydrogens were removed and re-added, and protonation states for the proton relay site were assigned[93]. Membranes were inserted using PACKMOL-Memgen[95] whereby the protein is embedded in a mixed bilayer patch composed of 1-palmitoyl-2-oleoyl-sn-glycero3-phosphoethanolamine (POPE) and 1-palmitoyl-2-oleoyl-sn-glycero-3-phosphoglycerol (POPG) in a 2:1 ratio, for a total of ~700 lipid molecules symmetrically distributed across the two bilayer leaflets (Supplementary Table 6). This composition reflects a simplified model chosen due to current limitations in available force fields for cyclopropanated lipids within the AMBER, which may affect the quantitative accuracy of our findings. The AMBER force field ff19SB[96] was used to represent the protein, the lipid21[97] parameters were used for the phospholipids, the OPC model was employed for water[98], and ions parameters were taken from Joung et al.[99]. A potassium ion concentration of 0.15 M was added, and the lipid packing was optimised with a 15 Å distance from the protein and a 17.5 Å water layer. Randomisation ensured proper lipid placement, and minimisation was performed post-generation. The system was built iteratively over 100 cycles with a convergence tolerance of 2.4 Å for optimal structural stability. *parmed* was used to repartition the mass of hydrogen atoms.

Each system was first subjected to a multistep structural relaxation via a combination of steepest descent and the conjugate gradient method using *pmemd.cuda* programme implemented in AMBER24 as described previously[57,100–104]. The system was then heated in two stages: first, from 0 to 100 K over 1 ns under constant-volume conditions applying harmonic restraints (k = 1 kcal·mol$^{-1}$·Å$^{-2}$) to the heavy atoms of both the protein and the lipids, subsequently, temperature was increased from 100 to 310 K under constant pressure (1 atm) for 1 ns, with harmonic restraints (k = 2 kcal·mol$^{-1}$·Å$^{-2}$) applied to the Cα atoms of the protein and *z* coordinates of the phosphorous atoms (P31) in lipids. This approach allowed for membrane rearrangement during the heating process. The system was equilibrated in a series of six equilibration steps at 1 fs time steps for 250 ps each (total 1.5 ns) with restraint on the protein coordinates under isotropic pressure scaling using the Berendsen barostat, whereas a Langevin thermostat (with a collision frequency of 1 ps$^{-1}$) was used to maintain a constant temperature. Initial equilibration was confirmed by stable temperature (~ 310 K), water density (~ 1 g/cm$^3$), and backbone RMSD (< 2 Å). Starting structures for production runs were chosen via average-linkage clustering (three clusters corresponding to the number of simulation replicas) of the equilibrated trajectory using *cpptraj*. Finally, production simulations were then conducted at 310 K with a 4 fs time step employing Langevin thermostat and anisotropic pressure scaling (under an isothermal-isobaric ensemble), with outputs recorded every 25,000 steps (100 ps). The Particle mesh Ewald algorithm was used to evaluate long-range electrostatic forces with a non-bonded cut-off of 9 Å. For each system (MdtF$^{WT}$, MdtF$^{V610F}$, and AcrB), three independent replicates were performed, each for 1 μs, resulting in a total simulation time of 3 μs per system. Trajectories post-processing was performed with *cpptraj* (Supplementary Fig. 22). The contacts at the PC1-PC2 domain by Cα atoms for all three protomers from the cumulative trajectories were calculated using "*nativecontact*" command (skipping native contacts) with a cut-off of 10 Å[105]. Extreme structures of maximum or minimum contacts (correspond to closing and opening of cleft) each associated with their respective minimum and maximum distances were chosen. The hbonds were calculated using "hbond" command with a distance cutoff 3.0 Å; angle cutoff 135°. The cut-off distance for salt bridge interaction was 3.2 Å between O and N atoms, calculated using *cpptraj*.

Hydration of the transmembrane (TM) region for each protomer within 10 Å of the proton relay site (PRS) was analysed using the grid tool in AMBER24, calculating the average water density (cut-off > 0.25) along the cumulative trajectory. Densities were normalised to bulk water (1 g/mL), and significant hydration sites (corresponding grid-points) were identified. Dynamic exchange was assessed by tracking water molecules that entered within 5 Å of the PRS in at least 1% of trajectory frames per monomer. Their single water trajectories were analysed within a 30 Å of PRS.

## Differential scanning calorimetry

Conformational stability was assessed by differential scanning calorimetry (DSC). MdtF$^{WT}$ and MdtF$^{V610F}$ were prepared at 7.5 μM, ± 75 μM R6G (10 x molar excess) in a 96-well plate, centrifuged at 4000 × *g* for 5 min to remove air and loaded onto an automated MicroCal VP DSC (Malvern Panalytical, Great Malvern, UK). A temperature gradient from 10 to 100 °C, at a scan rate of 1 °C/min was performed, using a pre-scan thermostat of 15 min, a filtering period of 5 s and in passive feedback mode. Scans were automatically buffer subtracted, concentration corrected, and a manual baseline set and subtracted in the MicroCal PEAQ-DSC software v1.64.

## Circular dichroism (CD) spectroscopy

All CD spectra were measured in 20 mM sodium phosphate, 150 mM sodium fluoride, pH 7.4 at a final concentration of 1 mM. Far-UV CD spectra were recorded on an Chirsacan CD spectrophotometer (Applied Photophysics, Leatherhead, UK), from 250 to 190 nm, with a step size of 0.5 nm, a bandwidth of 1 nm and a cuvette pathlength of 10 mm. A continuous temperature ramp from 20 to 90 °C, at 2 °C intervals at a rate of 0.23 °C/min was applied. The transition mid-points (Tm) of the spectra were analysed using a multi-wavelength fitting algorithm in Global 3 Analysis Software (Applied Photophysics, Leatherhead, UK). Scans were averaged, buffer subtracted and converted to mean residue ellipticity [θ] according to;

$$[\theta] = \frac{MRW \times CD}{10 \times l \times C}$$

Where [θ] is the mean residue ellipticity (deg cm$^2$ dmol$^{-1}$), CD is the raw CD value (mdeg), MRW is the mean residue weight (g/mol), *l* is the cuvette pathlength (cm), *C* is protein concentration (g/L). Spectra were plotted in GraphPad Prism 10.

## Reporting summary

Further information on research design is available in the Nature Portfolio Reporting Summary linked to this article.

## Data availability

Density maps and structure coordinates have been deposited in the Electron Microscopy Data Bank (EMDB) and the Protein Data Bank (PDB) with the following accession codes: apo-MdtF$^{WT}$ (EMD-53281 and PDB 9QPR), apo-MdtF$^{V610F}$ (EMD-53282 and PDB 9QPS), R6G-MdtF$^{V610F}$ (EMD-53283 and PDB 9QPT). MD simulation trajectories and the docking poses at the DBP, CH1, CH2 and CH3 are available at Zenodo: https://doi.org/10.5281/zenodo.15038634. GC-MS data is available at the NIH Common Fund's National Metabolomics Data Repository (NMDR) website, the Metabolomics Workbench, https://www.metabolomicsworkbench.org where it has been assigned Project ID PR002568. The LC-MS data generated in this study have been deposited in the MetaboLights database under accession code MTBLS13050. Source data are provided in this paper.

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

## Acknowledgements

M.A. and A.V.V. gratefully acknowledge the "One Health Basic and Translational Research Actions addressing Unmet Needs on Emerging Infectious Diseases (INF-ACT)" foundation by the Italian Ministry of University and Research, PNRR, mission 4, component 2, investment 1.3, project number PE00000007 (University of Cagliari) and the NIH/NIAID grant no. R01AI136799. The studies at King's College London and the University of Southampton were supported by a UKRI Future Leaders Fellowship (MR/S015426/1 and MR/X009580/1) to E.R. and a BBSRC iCASE studentship with UCB Pharma (BB/T008709/1) to R.L. The studies at the University of Southampton were also supported by a BBSRC industry studentship to L.S. (BB/T008768/1). The studies at the University of Oklahoma, USA, were supported by the NIH/NIAID grant RO1AI132836 to H.I.Z.

## Author contributions

R.L., H.I.Z., A.V.V., Z.A. and E.R. designed the project; R.L. and C.A. performed cryo-EM experiments; R.L. and J.S. performed cryo-EM analysis; R.L. and E.R. performed structural investigation and analyses; R.L. and O.D. performed biophysical experiments and analysis; R.L. cloned, purified, and characterised all protein constructs; M.R.U. and H.I.Z. performed bacterial accumulation assays and analysis; R.L. performed bacterial efflux assays and analysis; R.L. and L.S. performed bacterial susceptibility assays and analysis; R.L. performed lipid extraction and GC-MS; S.L. performed lipid LC-MS experiments and analysis; M.A. and A.V.V. carried out molecular docking and MD simulations and post-MD analyses; C.W.K., N.P., C.P., D.M., H.I.Z., A.V.V., Z.A. and E.R. supervised and/or financially supported the project; R.L., M.A., H.I.Z., A.V.V. and E.R. wrote the manuscript with input from the other authors.

## Competing interests

The authors declare no competing interests.
