## [Transparent Peer Review file · Nature Communications]

Molecular basis for multidrug efflux by an anaerobic-associated RND transporter

Corresponding Author: Dr Eamonn Reading

Editorial Note: Parts of this Peer Review File have been redacted as indicated to remove third party material where no permission to publish were obtained.

Version 0:

Reviewer comments:

Reviewer #1

(Remarks to the Author)

This is an interesting manuscript that provides 3.5 Å resolution cryo-EM structures of the MdtF RND efflux pump from *E. coli* that is induced during anaerobic metabolism. This transporter is a homologue of AcrB, which is currently the most comprehensively characterised RND efflux pump. In this manuscript, the authors appear to propose that MdtF operates using a different structural mechanism to AcrB. The manuscript describes three cryoEM structures of MdtF, the Substrate free (apo-MdtFWT), Substrate free (apo-MdtFV610F), and Substrate bound (R6G-MdtFV610F). The authors find that the conformational dynamics of the protein differ to that of AcrB in ways that are critical to maintain the integrity of the bacterium under anaerobic stress. The authors also characterise changes in the lipidome under anaerobic stress, and indicate key residues that may contribute to substrate promiscuity.

This work is of significance to the field. The manuscript is written to directly compare the characterisation of MdtF to AcrB.

However, key questions about the MdtF structures are not explicitly addressed by the authors, for example: 1) are these distinct? 2) How different are they to related proteins? 3) What is the significance of the V610F mutation? As outlined below in the Issues, there are other gaps in the data analysis that mean the conclusions are not adequately supported. Overall, the methodology in its current form is insufficient for the work to be reproduced, although this can easily be addressed.

Despite the issues outlined below, this work has the bones of a very nice manuscript, although I believe publication in its current form would be premature.

Issue 1)

The first issue is that the Introduction is both too short, and also rather chaotic and disjointed. The introduction contains dangling sentence fragments. It also contains very long, run-on sentences that should be split into two or three more concise sentences for clarity. The research findings of the manuscript rely heavily on the comparison of MdtF and AcrB, however there are many places where the writing style is ambiguous and contextually unclear. There are references to 'the protein' or 'the system' without explicitly stating whether 'the protein' being referred to is MdtFWT, MdtFV610F or AcrB. Another example is "The membrane is remodelled" is ambiguous. Which membrane? Inner? Outer? Both? This ambiguity makes the manuscript difficult to follow, to an extent that the manuscript does not appear to have a clear or coherent story. It's very difficult to tell whether this is because the research lacks the appropriate level of impact, or whether this is due to problems with the writing, such as omissions in critical background information or a lack of explanation of key results.

Issue 2)

The manuscript is written from the point of comparison of MdtF to AcrB. While AcrB has been comprehensively studied over many years, the authors write for an audience that is completely immersed in the RND efflux pump fields and has extensive experience in AcrB, and do not write for a general audience. This is quite off-putting, as for some sections of the manuscript key background information is often missing, while for other sections there is an extremely high level of detail present in the Introduction that is not referred to at all later in the manuscript.

For example, in the section on docking, the authors state that key residues differ between AcrB and MdtF. While they state what these residues are in MdtF, they do not describe what these residues are in AcrB. They assume that the AcrB

sequence is implicit knowledge. While the authors do provide a sequence alignment as Supporting Information, if the identity of the amino acid is important, it should be stated explicitly.

Issue 3)

The manuscript also contains inconsistent terminology. For example, page 6 states: "... protomer states sequentially cycling through three conformations, i.e., access (or loose), binding (or tight), and extrusion (or open) (hereafter A, B, and E, respectively).."

in fact, the conformations are never referred to as A, B, and E. They are only ever referred to as access, binding and extrusion, until it comes to the Methodology for Molecular Dynamics simulations where they are referred to as L, T and O, which is presumably Loose, Tight and Open. This is never explicitly explained.

Issue 4)

A key point in the manuscript is that the plasticity of the MdtF drug-binding domain. The authors state the V610F mutation is key to the description of substrate plasticity of the MdtF drug-binding domain. They use a V610F mutation to alter the substrate profile and solve the cryoEM structure for this MdtFV610F mutation. Despite going to great lengths in Figure 1 (PC1, PN1 etc) and Supplementary Figure 13 to identify key features of the drug-binding domain (such as the Switch Loop, Distal Loop, Flexible Loop, Hoisting Loop, etc), the authors do not inform the reader where V610F is located within the MdtF structure. I have assumed that this lies within the Switch Loop, however this is never explicitly stated, or indicated in a figure.

The authors state that the V610F point mutation "is induced by" antibiotic pressure and causes swelling of the distal binding pocket, which "reverberates" allosteric changes to the entire protomer to make it more "AcrB like", however this is an assertion and not explicitly supported by their research findings. For example, the authors do not actually show that V610F has altered efflux activity, nor do they discuss what is known about this mutation in the existing literature. What the authors mean by "AcrB-like" needs to be described in more detail. What do they mean when they say V610F was induced by antibiotic pressure? What do the authors mean by "reverberates" allosteric changes? This needs to be more precisely defined and supported by the MD simulations.

If the structures show a big conformational difference from V610F, and the assays show a big activity difference, that's quite exciting, however no quantitative measurements are provided in the manuscript. Docking energies are not a solid quantitative measurement.

Issue 5)

In the Introduction, a lot is made of the importance of cyclopropanation of inner membrane lipids which increase both membrane fluidity, rigidity and stability (the three together is unusual). This is deemed important enough to perform lipidomic analysis and identify the lipid composition of the MdtF SMA single particles. However, this was not discussed any more, and the role of the lipid composition in maintaining the PFM was not discussed.

Despite the importance of these lipids in protein function, MD simulations were conducted in a membrane composed of POPE and POPG in a 2:1 ratio. The conformational dynamics of the protein from MD simulations in 2:1 POPE:POPG is used as a proxy for the lipidomic analysis of a membrane containing (roughly) 80-90% PE membrane, with equal ratios of PG, cardiolipid membrane; and 50% saturated, 25% unsaturated and 25% cyclopropanated fatty acid tails. These membrane compositions are quite different.

Given the emphasis the authors place on the importance of the lipid composition, it is difficult to see how the conformational dynamics in a standard POPE/POPG membrane can give an accurate representation of the conformational dynamics in the cyclopropanated membrane lipidome.

The take-home points of the manuscript are focused on a comparison of the conformational dynamics from MD simulation and the cryo-EM structures. From the author's own statements, having the correct lipid environment is likely to impact the conformational dynamics of MdtF, yet the authors never link these differences in membrane lipid composition to the experimental results, and the reader is led to believe that the simulations being reported have been carried out in the experimental membrane lipidome.

Issue 6)

The manuscript also emphasizes the RMSD of 1-1.6Å between the corresponding Access, Binding and Extrusion states of ArcB and MdtF, seeming to emphasise that these are large transitions. In fact, these RMSDs show that the protein structures are extremely similar. Even RMSDs of 3 Å are within the expected conformational spread of the cryoEM structures.

The authors also state on Page 13 "although our MD simulations reveal relatively large structural departures in the MdtF dynamics compared to the cryo-EM geometries, the simulations of AcrB provided findings that are in good agreement with experimental data³⁴."

There is no evidence presented to show that the MD simulations have relatively large structural departures in the dynamics compared to the cryoEM geometries. This would require some kind of measurement of the conformational dynamics, etc, RMSD, RMSF, cluster analysis, etc. No RMSDs or other measure of global conformational similarity or difference has been presented. The authors also do not explicitly note that MD simulations are performed in a membrane that's very different from

the lipidome identified.

The authors describe transmembrane conformational changes and acid responsive transport mechanism not found in other RND efflux pumps, which they have identified from the cryoEM structure, MD simulations and functional assays. The key questions here are whether these are novel, whether the conclusions are convincing, and how the MD simulations and functional assays were performed.

The authors state that MdtF gives moderate efflux under neutral conditions. They speculate this may be related to maintenance of PMF under stress conditions. It is not clear why these ideas are linked? The rationale for this speculation needs to be set out more clearly.

The authors go on to state that under acidic stress, there is increased efflux efficiency to “battle xenobiotic attack and host-toxin clearance”. What is meant by efficiency? Energy efficiency? Increased activity?

Issue 7)

The authors over-claim their results without substantiating them. The crux of the authors findings (as it is currently written) is that they have cryoEM structures for MdtF, and they show that when MdtF is in a membrane that more closely represents a non-stressed E coli membrane, its conformational dynamics are slightly differently to AcrB. The manuscript basically says that MdtF is more rigid in the core and the peripheral regions of the protein are floppier.

The authors make very big claims about the distributions of point mutations for the 30% of the sequence that isn't conserved between AcrB and MdtF. The authors support this with Figure S10, which shows that mutations are spread throughout the protein. However, this comes across as though the authors are trying to re-invent 50 years of 'homologous proteins share the same fold', and this is also the case for every other globin, ion channel, membrane transport protein and enzyme that is known to biology.

Likewise, while it is important to point out that AlphaFold3 cannot build a machine learning model of any RND efflux pump in a trimeric A-B-E conformation, this is a known limitation of AlphaFold ML/AI models. This is an important point that should be stated. Please point it out, but cite all the other people who have also pointed this out! This should contribute to the body of literature, rather than fragmenting it. “Our study provides the first mechanistic view of how MdtF can recognise a wide range of harmful cellular and xenobiotic substances, incl. antibiotics”. The authors also state that this functional data will help with EPI design. I'm not sure I agree with that claim, but I think presenting the mechanistic basis of an altered function point mutation is worthwhile on its own merits.

Issue 8)

The figures generally need work.

Figure 1A and Figure 3 are generally unclear and do not support the written rationale adequately. The Position of V610 /F610 not clear from the simulations. In Figure 3A – you can't see where V610 is and where F610 is. This is important! The image with the hydrophobic nook has the larger side chain and the hydrophobic nook is not apparent in the figure. Residues should be coloured differently in part A to highlight them. Even the position of these residues is unclear from the figure. I would have thought that V610 was part of the switch loop. If it is, please state this explicitly.

Figure 3B – fluorescence polarisation is not explained in the main text or the methods. This needs to be clarified for a non-expert audience.

Figure 2C – Supplementary Figure 25 is so much easier to understand than Fig 2C because of the use of transparency in S Fig 25.

Figure 4 needs further explanation. The significance of A675, S676 is not explained adequately in either the figure caption or text.

Figure 5A doesn't actually show you where the proton relay switch is, however the top view of Figure 5C is nice. Figure 5D gives the H-bond occupancy, however this is not a standard format for a H-bond occupancy plot and it's not stated in the text or figure legend what the box plots mean! These are not well explained and they are not presented in a standard format that is easy to interpret.

Gels and micrographs in figure S1 poor image quality, low resolution, jpeg artefacts and fringing, however we acknowledge this could that be a problem with the proofs.

Figures S11 S12 have inadequate labelling. Can't tell the degree to which visual differences between protomers in S11 are due to angle. Pairwise RMSD would likely support the author's point.

Supp Fig 12 states that AF3 was not successful, however the models look the same by visual inspection. Also the colour key is not given, so you can't tell which is which. Again, a pairwise RMSD would show the AF prediction monomers are the same while the structural monomers are distinct, which would further support the authors' important point about the importance of structural data over AlphaFold predictions. While this is done for AcrB and MdtF in Figure 2B, it would be great to have it for the AF predictions as well.

Supp Fig 26 – the whole argument till now is about the binding pocket, now we're switching to channel 1 channel 2 and channel 3.... Please link these clearly in the main text if it's important. Supp Table 3 – docking free energies. How reliable are these? Connection back to the general idea of how this works together with everything else is not clear.

Issue 9)

Why were the simulations done in KCI?

How were protonation states assigned? Were pKa values for residues predicted with some tool such as a PropKa, or were protonation states assigned based on canonical pKa values? The methods state that residues were protonated at physiological pH, but do not indicate the physiological pH relevant to their system.

Reviewer #2

(Remarks to the Author)

This manuscript describes the first cryo-EM structure of the RND-type efflux pump MtdEF from *E. coli* that is expressed during early stationary phase growth and in response to environmental stress, such as low pH. The authors provide interesting evidence for significant differences in the tertiary structure of MtdF, as compared to AcrB, which reflect its distinct role during stationary phase and as a stress response mechanism. The structural data is augmented by MD simulations and wet lab data. For the most part, the data support the conclusions of the authors. Since MtdF has not been as extensively studied as the constitutively-expressed AcrAB pump, the manuscript would provide a valuable contribution to the field. However, there are several issues that need to be addressed before publication.

General comments.

Since the description of the cryo-EM structures in the first half of the results sections contains detailed descriptions of the three functional conformations of the MtdF monomers and their conformation transitions, it would be helpful for the reader who is not an expert on RND-type pumps to include a paragraph in the Intro needs more background information on RND structure and function.

The Introduction and the subsections of the Results describing the cryo-EM results (including the figure legends) and discussion would benefit from extensive editing for grammar, clarity, and accuracy. Some of the descriptions of the key findings are cursory, and would be difficult for a non-expert to follow.

Most of the data cited in the main manuscript is in the supplemental data section (31 figures and 4 tables). Some of these data are important for supporting the conclusions of the paper. It was tedious to have to refer to the supplemental data 34 times (one of the tables was not mentioned in the text). Most readers will not bother to look at the supplemental data. I suggest that the authors reevaluate the data and move the figures that are key to supporting the main conclusions to the main paper.

The axis labels on the supplemental figures that show the results of laboratory experiments are too small to read. Please increase the size of the figure and font size.

Some specific comments related to the general comments.

Page 7. The authors use the phrase "promiscuous substrate efflux" to describe that activity of MdtF. However, the term "polyspecific binding or efflux" site provides a more accurate description, as the binding site accommodates substrates with a set of general chemical properties but does not bind to everything.

Page 8. The FP data indicates that MdtF binds to Rhodamine. Therefore, it is curious that the authors were unable to obtain a cryo-EM structure of Rhodamine bound to WT MdtF. The authors should address this discrepancy. Is it possible that the FP assay is not measuring specific binding to the substrate binding site?

The authors state that "Surprisingly, we found that the exponentially growing and the stationary cells producing MdtEF differ in the accumulation of HT with the stationary cells accumulating significantly lower amount of HT than the exponential cells (Fig. 6A and Supplementary Fig. 23A)." This data is in Supp Fig, 23B. However, the difference between experiments shown in 6A and supp fig. 23B is not clear. They are described as essentially the same experiment, but the results are very different.

Supp Fig 23A, which needs labels for each protein, shows pump expression is the same in exponential and stationary cells. However, the lack of expression of the pore in stationary phase leaves the question of altered membrane permeability unanswered. The observation that HT accumulates to the same level in exponentially growing and the stationary cells with vector only (even without pore), suggesting that permeability of HT is the same in exponential and stationary phase cells. Therefore, decreased HT accumulation in stationary phase cells could be caused by increased pump activity in stationary phase. Since membrane composition markedly changes in stationary phase, it seems possible that the activity of MtdF, an integral membrane protein, may be responsive to the altered membrane composition.

Fig 6B. As a control, it would have been useful to include AcrAB in this experiment.

Figs.1A and 6C. The figures shows that RND substrates are extruded from cytoplasm. Since the substrate-binding site of MtdF is in the periplasmic space, substrates are extruded from the periplasmic space. I suggest editing the figures accordingly.

The MIC data shown in Supp table 4 is not mentioned in the text. In addition, the table does not contain susceptibility data no MICs for the V610F mutant. While this data exists in the literature, it would be useful to include MIC data for a broad range of

substrates in this manuscript to verify the phenotype of the WT and V610F mutant pumps using the engineered strains described in the manuscript.

Page 16. Please note that the sentence containing “such as the ESKAPE (Enterococcus faecium, Staphylococcus aureus, Klebsiella pneumoniae, Acinetobacter baumannii, Pseudomonas aeruginosa, and Enterobacter spp.) classes” should be revised, as the Gram-positive pathogens in this list do not encode RND-type pumps.

Reviewer #3

(Remarks to the Author)

The manuscript submitted by Lawrence and colleagues describes the molecular characterisation of MdtEF, a proton motive force-driven efflux pump from the resistance-nodulation-cell division superfamily. The manuscript is overall well-written, is easy to understand and is innovative in the findings presented.

I have reviewed the lipidomics aspects of the manuscript. The authors should consider the following feedback.

1. The GC-MS and LC-MS raw and processed data should be made publicly available in a metabolomics data repository (Metabolights or Metabolomics Workbench). The output data should also be included in supplementary information for readers to review.
2. The GC-MS/FAMES extraction method is appropriate. The GC-MS assay has a short analysis time which would not support full chromatographic resolution of all fatty acids. The authors need to provide evidence that all fatty acids reported are identified correctly with high confidence. A commercial mixture of bacterial acid methyl esters was used for identification of fatty acids but no information on the source or composition of this mixture was provided and should be provided. Also, were all detected FAMES from the sample present in the commercial mixture including cyclopropanated fatty acids or were some detected fatty acids in the samples not present in the commercial mixture? If the latter then how were those fatty acids detected but not in the commercial mixture identified? Full methods are required.
3. GC-MS/FAMES assay. Can the distribution of carbon number be reported as this will also be of biological relevance/importance.
4. The LC-MS assay appears to be appropriate. However, the annotation/identification of lipids is not performed appropriately. Identification has been performed using accurate mass only which provides probable errors. Why were MS/MS data not collected and used in the identification process? Many of the lipids reported have a mass which can also be linked to another lipid (e.g. PC and PE species as well as PG and PS species). Without further inclusion of methods applied in Lipidmaps including m/z error, adduct type etc and further experimental evidence, lipid identifications should not be reported.
5. LC-MS assay. The parameters used by the software and the strategy applied for processing of the data should be reported. Were data processed to search for specific m/z values in a targeted way or was an untargeted strategy applied? Full information should be included.

Reviewer #4

(Remarks to the Author)

Reviewer #5

(Remarks to the Author)

Escherichia coli multidrug transporter MdtF is a special resistance-nodulation-cell division (RND) transporter because it is upregulated and functional in low oxygen, extreme acid and nutrient starvation conditions. In this manuscript, Lawrence et al. solved the cryo-EM structures of Apo-MdtF WT, Apo-MdtF V610F and R6G-MdtF V610F within native-lipid nanodiscs. The thorough comparison of the asymmetric structures of MdtF and its close homolog AcrB, in addition to functional assays and substantial computational analysis such as molecular dynamics simulations and docking, reveals the structural and functional specialties of MdtF. However, the linkage between structural findings and the pump function particularly in extreme conditions is somewhat insufficient.

There are a few major concerns to be addressed:

1. The structural analysis mainly focused on Apo-MdtF WT, which is overall alike to AcrB. The structures of Apo-MdtF V610F and R6G-MdtF V610F were not described or compared in enough details. Are these two structures also asymmetric and alike? It is confusing since the author stated that “MD simulations reveal that the V610F mutation induces ..., resulting in a somewhat more symmetric MdtF conformation ... (P11)”. Why was only one R6G ligand identified in the substrate-bound V610F structure (9QPT)? In which protomer of the R6G-MdtF V610F structure was the bound R6G identified? For the analysis and MD, it was often unspecified which chain of Apo-MdtF V610F or R6G-MdtF V610F was used.
2. As a major anaerobic RND transporter, nitrosyl indole derivatives are the important substrates for MdtF. The experimental structure of MdtF in complex with nitrosyl indole derivatives is more meaningful. Why was only R6G used as the ligand in the complex structure determination? The binding affinity estimation of MdtF WT and V610F by docking (Supplementary table 3) suggested that the CH3 binding might be lightly stronger for nitrosyl indole than the DBP binding. It is possible that the

nitrosyl indole derivatives are favored by the different binding site in MdtF compared to R6G in MdtF or AcrB. In addition, it is more convincing to provide experimentally determined binding affinities of MdtF.

3. All the cryoEM structures, MD simulations and most of functional assays were conducted at physiological pH. However, MdtF is known as an acid-responsive transporter in some extreme conditions. The identified proton relay residues of MdtF were highly conserved among RNDs. Thus, these results didn't provide sufficient evidences for the acid-responsive speciality of MdtF. In addition, from the NPN efflux assay, it was shown MdtEF confers enhanced activity at lower pH. This result is intriguing. Identifying the key residues contributing to this unique property of MdtF would help to elucidate the special acid-responsive mechanism that may distinguish MdtF from AcrB and many other RNDs.

Some minor comments are listed below that may improve the clarity and readability of the article.

1. In the Fig 3, location of the V610F mutation site and DBP in the porter domain is obscure. The protein surface (panel A) could be colored according to the hydrophobicity scheme.
2. The Supplementary Fig 9 could be labeled or marked with secondary structural elements, domains and essential residues.
3. Related to Supplementary Fig 12, although it was stated the asymmetric structure of MdtF WT was not captured by AlphaFold 3 prediction, it is not clear which state of the protomers the AF predicted monomer (or trimer) resembles.
4. The assignment of each monomer state of MdtF WT was based on the RMSD calculations of the structural comparison with AcrB. It is not clear if the only C alpha atoms were counted in the calculation for each monomer. As to the Chain B of MdtF WT, the RMSD values for "Access" (1.6 Å) and "Binding" (1.9 Å) are very close but obviously less significant than the best RMSD for the other two chains (Chain A, 1.0 Å; Chain C, 1.1 Å). The authors claimed Chain B as the "binding-like access R2" state and reported some major TM movement of this "Access" state. It is not clear how similar their porter domains are when compared with AcrB. Regional comparison and RMSD calculation may help to explain what structural variance may cause the "high" RMSDs.
5. In the Supplementary Fig 13A, it is not clear what state of protomers was presented.
6. P11, "there were more distinct intermediate and extreme conformations allowed by (3 vs. 2 and 2 vs. 1 in the access and binding states, respectively, Fig. 4B)", what exactly do "3 vs. 2 and 2 vs. 1" refer to?
7. In the Supplementary Fig 20, the contour levels for the map densities of the proton relay residues were not specified. Please also note the cryoEM map is not calculated based on electron densities.
8. In the Supplementary Fig 28-31, it is better to show the site of V610F to mark the main structural difference between WT and V610F. The docking scores or energies should be provided in addition.
9. There are numerous typos in the supplementary figures, eg, Fig S6, S17, S20, S22, S24, etc.
10. The blotting figures are too small in the Supplementary Fig 21, 26C.

Reviewer #6

(Remarks to the Author)

Laurence and colleagues report on the structure-function relationship of MdtEF, a RND multidrug transporter overexpressed under anaerobic or stress conditions. To understand the particularity of MdtF transporter, they decided to compare it with the constitutively expressed transporter AcrB. The methodology developed in the ms is based on a structural method, cryoEM combined with molecular dynamics, and functional assays. Consistent structural and functional results were obtained providing clue to better understand how MdtF functions.

General comments

Solid work has been carried out to explore the molecular basis of MdtF function. On one side, the structure of MdtF wt and mutant with and without ligand were determined at neutral pH, with native lipids using the SMA polymer. MdtF wt exhibits structural differences from AcrB. Mutation experiments in binding site and MD simulation seem to suggest that MdtF activity can be modulated by changing the binding site accessibility. On the other side, functional analysis indicates that external pH affects MdtEF activity. Its efficacy is increased at low pH. This raises an important question. What is the conformation of MdtF at low pH? Is there a change for the binding site? This piece of the puzzle is missing to fully understand the role of pH in the functioning of MdtF.

The authors proposed that the MdtF identity is defined by its proton relay that could be a "truly distinct mechanism of coupling between proton translocation and substrate export in MdtF" (p13). It is not clear how to correlate the observation in the computational experiment that R969 is directly involved in the proton relay and why this would have an impact on substrate export. This point needs to be clarified. Moreover, it is not clear how the proton relay would interplay with the pH-dependent activity (in the paragraph "MdtF drug transport is responsive to proton load"). Indeed, it is important for the reader to have a clear idea of what makes MtdF specific: is it specific substrates that are

produced under extreme conditions that need to be expelled or is it the conditions themselves that enable the function of MdtF?

Experiments with the V601F mutant suggest that it is possible to change substrate specificity. There is a slight increase in the affinity of R6G, making it possible to resolve the structure of the R6G-MdtF complex (V601F). This mutation appears to confer a function similar to that of AcrB at neutral pH. It would be interesting to assess whether the activity of MdtF (V601F) is also pH-dependent like MdtF wt.

Molecular basis of MdtF efflux has been studied at low pH and not under anaerobic conditions, I would suggest changing the title of the ms accordingly

Additional comment

Figures 1A and 6C appear redundant. I suggest to group them in one figure..

Version 1:

Reviewer comments:

Reviewer #1

(Remarks to the Author)

The authors have revised their manuscript extensively and have addressed the majority of the reviewer comments. I have two minor comments that should be addressed before publication:

1) Figure 5D: what do the circles above and below the bars on the bar graph mean? Presumably they represent errors, but this should be explicitly stated.

2) There are still a few grammatical and typographic errors that need to be corrected.

Reviewer #2

(Remarks to the Author)

The authors have done a tremendous job in responding the reviewer's comments. As a result, the manuscript is greatly improved and was a pleasure to read. This is an important paper and should be published. This new and improved version will make these interesting results accessible to both experts and non-experts.

Reviewer #3

(Remarks to the Author)

The authors have provided a revised manuscript based on reviewer's feedback. For reviewer 3 some of the feedback has been applied appropriately in the revised manuscript though other feedback has not. The authors need to consider the following.

1. The GC-MS data have been included in a metabolomics data repository (Metabolomics Workbench). However, the LC-MS data has not been included in a data repository but should be.
2. The authors have included a GC-MS chromatogram of the BAME mixture and the sample in supplementary information which is appropriate. However, my feedback for version 1 of the manuscript was to define with confidence the fatty acid methyl ester identifications. This has not been performed. For example the BAME mixture has 26 components but the BAME chromatogram has only 21 peaks, where are the peaks for the other 5 BAMEs? I suspect there is co-elution. Also, how do the authors know which BAME relates to which peak with a specific RT in a mixture of BAMEs, surely this would require analysis of each BAME separately? Further information is required.
3. The part number of the BAME mixture should be included in the methods section.
4. The LC-MS data processing software parameters should be included as recommended previously.

Reviewer #4

(Remarks to the Author)

Reviewer #5

(Remarks to the Author)

This revised manuscript has been exceptionally improved. All my previous concerns have been adequately addressed. I have no further comments. Although the detailed mechanisms on how MdtF functions under anaerobic conditions need further dissection, the current findings reported in this manuscript are interesting and presumably supplement the mechanistic understanding of RND transporters in general.

Reviewer #6

(Remarks to the Author)

The authors have provided a detailed rebuttal to the points raised by the reviewers and changed the text of the manuscript. The responses are satisfactory and the revised manuscript is improved. I consider the data worth publishing

REVIEWER COMMENTS

Reviewer #1 (Remarks to the Author):

This is an interesting manuscript that provides 3.5 Å resolution cryo-EM structures of the MdtF RND efflux pump from *E. coli* that is induced during anaerobic metabolism. This transporter is a homologue of AcrB, which is currently the most comprehensively characterised RND efflux pump. In this manuscript, the authors appear to propose that MdtF operates using a different structural mechanism to AcrB. The manuscript describes three cryoEM structures of MdtF, the Substrate free (apo-MdtFWT), Substrate free (apo-MdtFV610F), and Substrate bound (R6G-MdtFV610F). The authors find that the conformational dynamics of the protein differ to that of AcrB in ways that are critical to maintain the integrity of the bacterium under anaerobic stress. The authors also characterise changes in the lipidome under anaerobic stress, and indicate key residues that may contribute to substrate promiscuity.

This work is of significance to the field. The manuscript is written to directly compare the characterisation of MdtF to AcrB.

However, key questions about the MdtF structures are not explicitly addressed by the authors, for example:

1) are these distinct?

***E. coli* MdtF, AcrB and AcrF are all on the same phylogenetic clade (see diagram adapted from [10.1101/2024.11.22.624703](https://pubmed.ncbi.nlm.nih.gov/4111101/) - presenting a correlation coefficient-analysis map of pairwise sequence alignment [REDACTED]).**

AcrF does not yet have any experimentally derived structural information available . Due to these reasons and the juxtaposition of MdtF having high sequence similarity to the constitutively expressed AcrB, but being distinctly upregulated in anaerobic and sessile conditions, we chose to compare *E. coli* MdtF and AcrB in detail throughout the paper. We have now noted this more prominently in the manuscript.

2) How different are they to related proteins?

We have now provided further, Ca-atom RMSD analysis in Figure 2 and associated Supplementary Table 7. We compared global RMSD for MdtF protomer states to known RND transporter structural models for AdeB, AcrD, AcrB, OqxB, MexB and CusA. This revealed MdtF asymmetric state to be most like that of AcrB but found its access state to be distinct across homologs. New localised Ca-atom RMSD analysis helped to better portray the significant differences between MdtF and AcrB asymmetric states, especially in the transmembrane domains of the access state. We also compare MdtF structures to AcrB structural models in both detergent micelles and SMALPs, and find this observation to be consistent across both environments (Supplementary Figure 12). We thank the reviewer for this suggested analysis; this has significantly improved the manuscript and confidence in the conclusions made.

3) What is the significance of the V610F mutation?

To clarify the importance of the V610F mutation in the manuscript we have included expanded explanations on its emergence in the introduction and insights from our structural models and molecular dynamics analysis in the results

sections. To supplement this, we have also conducted MIC measurements on this single point mutant in comparison to WT MdtF to corroborate these changes to the resistance phenotype and confirm the changes in the $\Delta 9$ -Pore *E. coli* strain which eliminates issues relating to substrate diffusion through the outer membrane.

As outlined below in the Issues, there are other gaps in the data analysis that mean the conclusions are not adequately supported. Overall, the methodology in its current form is insufficient for the work to be reproduced, although this can easily be addressed.

Despite the issues outlined below, this work has the bones of a very nice manuscript, although I believe publication in its current form would be premature.

Issue 1)

The first issue is that the Introduction is both too short, and also rather chaotic and disjointed. The introduction contains dangling sentence fragments. It also contains very long, run-on sentences that should be split into two or three more concise sentences for clarity. The research findings of the manuscript rely heavily on the comparison of MdtF and AcrB, however there are many places where the writing style is ambiguous and contextually unclear. There are references to 'the protein' or 'the system' without explicitly stating whether 'the protein' being referred to is MdtFWT, MdtFV610F or AcrB. Another example is "The membrane is remodelled" is ambiguous. Which membrane? Inner? Outer? Both? This ambiguity makes the manuscript difficult to follow, to an extent that the manuscript does not appear to have a clear or coherent story. It's very difficult to tell whether this is because the research lacks the appropriate level of impact, or whether this is due to problems with the writing, such as omissions in critical background information or a lack of explanation of key results.

We have now addressed this by being more precise with our language when referencing proteins or membranes in the manuscript. We have also made additions to the introduction of the manuscript to make it more coherent and easier for the reader to follow. Our introduction and discussion have now also been re-worked to better highlight the key results.

Issue 2)

The manuscript is written from the point of comparison of MdtF to AcrB. While AcrB has been comprehensively studied over many years, the authors write for an audience that is completely immersed in the RND efflux pump fields and has extensive experience in AcrB, and do not write for a general audience. This is quite off-putting, as for some sections of the manuscript key background information is often missing, while for other sections there is an extremely high level of detail present in the Introduction that is not referred to at all later in the manuscript.

We have reworked the introduction sections of the manuscript to bring in more of the relevant background suggested so as to better cater to a more general audience. We also have attempted to better link the detail in the introduction to other areas of main text.

For example, in the section on docking, the authors state that key residues differ between AcrB and MdtF. While they state what these residues are in MdtF, they do not describe what these residues are in AcrB. They assume that the AcrB sequence is implicit knowledge. While the authors do provide a sequence alignment as Supporting Information, if the identity of the amino acid is important, it should be stated explicitly.

As requested, we have included the comparative annotation of sequence alignment of channels (used in the docking for MdtF) and binding site residues of MdtF and AcrB in Supplementary Fig. 9b. We also now explicitly refer to this picture in the Methods section of the revised manuscript.

Issue 3)

The manuscript also contains inconsistent terminology. For example, page 6 states: “... protomer states sequentially cycling through three conformations, i.e., access (or loose), binding (or tight), and extrusion (or open) (hereafter A, B, and E, respectively).”

in fact, the conformations are never referred to as A, B, and E. They are only ever referred to as access, binding and extrusion, until it comes to the Methodology for Molecular Dynamics simulations where they are referred to as L, T and O, which is presumably Loose, Tight and Open. This is never explicitly explained.

We have rectified this by making sure the references to the access, binding, and extrusion protomers are consistent throughout the manuscript, thereby removing any reference to Loose (L), Tight (T), or Open (O).

Issue 4)

A key point in the manuscript is that the plasticity of the MdtF drug-binding domain. The authors state the V610F mutation is key to the description of substrate plasticity of the MdtF drug-binding domain. They use a V610F mutation to alter the substrate profile and solve the cryoEM structure for this MdtFV610F mutation. Despite going to great lengths in Figure 1 (PC1, PN1 etc) and Supplementary Figure 13 to identify key features of the drug-binding domain (such as the Switch Loop, Distal Loop, Flexible Loop, Hoisting Loop, etc), the authors do not inform the reader where V610F is located within the MdtF structure. I have assumed that this lies within the Switch Loop, however this is never explicitly stated, or indicated in a figure.

We have now updated Figure 3 to demonstrate the location of the V610/F610 residue in MdtF and have complemented this in the manuscript text.

The authors state that the V610F point mutation “is induced by” antibiotic pressure and causes swelling of the distal binding pocket, which “reverberates” allosteric changes to the entire protomer to make it more “AcrB like”, however this is an assertion and not explicitly supported by their research findings. For example, the authors do not actually show that V610F has altered efflux activity, nor do they discuss what is known about this mutation in the existing literature.

What the authors mean by “AcrB-like” needs to be described in more detail. What do they mean when they say V610F was induced by antibiotic pressure? What do the authors mean by “reverberates” allosteric changes? This needs to be more precisely defined and supported by the MD simulations.

If the structures show a big conformational difference from V610F, and the assays show a big activity difference, that’s quite exciting, however no quantitative measurements are provided in the manuscript. Docking energies are not a solid quantitative measurement.

We believe this point is also partly addressed by our response to ‘key question 3’ above. As mentioned, we have now explicitly stated the significance of the V610F mutation in the literature. In addition to this, we carried out additional MIC experiments on this single point mutant to further corroborate the effect of this mutation (this now data presented in Figure 3). The comment relating to ‘AcrB-like’ has been removed due to its ambiguity. In addition to this we have also made changes in the introduction and results to clarify in more detail the changes that we observed to overcome other ambiguous language in the manuscript.

Issue 5)

In the Introduction, a lot is made of the importance of cyclopropanation of inner membrane lipids which increase both membrane fluidity, rigidity and stability (the three together is unusual). This is deemed important enough to perform lipidomic analysis and identify the lipid composition of the MdtF SMA single particles. However, this was not discussed any more, and the role of the lipid composition in maintaining the PFM was not discussed.

Despite the importance of these lipids in protein function, MD simulations were conducted in a membrane composed of POPE and POPG in a 2:1 ratio. The conformational dynamics of the protein from MD simulations in 2:1 POPE:POPG is used

as a proxy for the lipidomic analysis of a membrane containing (roughly) 80-90% PE membrane, with equal ratios of PG, cardiolipid membrane; and 50% saturated, 25% unsaturated and 25% cyclopropanated fatty acid tails. These membrane compositions are quite different.

Given the emphasis the authors place on the importance of the lipid composition, it is difficult to see how the conformational dynamics in a standard POPE/POPG membrane can give an accurate representation of the conformational dynamics in the cyclopropanated membrane lipidome.

The take-home points of the manuscript are focused on a comparison of the conformational dynamics from MD simulation and the cryo-EM structures. From the author's own statements, having the correct lipid environment is likely to impact the conformational dynamics of MdtF, yet the authors never link these differences in membrane lipid composition to the experimental results, and the reader is led to believe that the simulations being reported have been carried out in the experimental membrane lipidome.

We have expanded our introduction to include a better explanation of the lipid composition and the enrichment of CFA lipids in its expression conditions. However, we need to clarify that we did not characterise changes in the lipidome under anaerobic stress and stationary phases (as others have done this previously) and what we set out to do here was to only characterise the phospholipid types of our MdtF SMALP samples. This analysis confirmed that PE was the predominant lipid type and that SFA, UFA and CFA fatty acids were present in the phospholipid extracted. To rectify this, the language relating to CFA identification has been modified in the manuscript to more explicitly state this. Also, a re-worked discussion aims to better describe a potential balance between MdtF efflux activity and PMF-related fitness cost, whilst better expressing that more work is needed to be done to explicitly understand this relationship (not only for MdtF but for the RND efflux class in general).

The reviewer is right in highlighting the differences in the composition of the membrane between the characterised SMALPs and the computational setups. This is due to the lack of well-validated all-atom force field tailored to cyclopropanated fatty acids (particularly within the AMBER lipid force field family used in this work), at opposite with unsaturated and saturated ones used in our simulations. In addition, the precise stereochemical composition of cyclopropanated fatty acids (e.g., R/S configurations, cis/trans isomers, unsaturated/saturated/Cp distribution for each specific PE/PG headgroups) is not well characterized, yet it can influence the dynamics (J. Phys. Chem. B 2015, 119, 17, 5487–5495). This present further

challenges in accurately modelling MdtF in a model membrane, as introducing cyclopropanated fatty acids without accurate details could compromise the reliability of MD simulations.

Nonetheless, it has been reported that cyclopropanated fatty acids (in DPPC derivatives) increase the fluidity of the membrane (*J. Phys. Chem. B* 2015, 119, 17, 5487–5495; simulations performed using the GROMOS force field, not the AMBER force field we employed here). Thus, it is reasonable to suppose that using a more realistic membrane would enhance the differences in the structural dynamics reported in the original manuscript. In this respect, the reviewer is right in noting changes in the composition of the membrane can affect fluidity, rigidity and stability in different ways. This would make for an exciting future study which builds on the work presented here on the first high-resolution experimentally derived structural information for the MdtF pump. We have included detail on these future directions in the discussions section of the manuscript.

Following the reviewer's criticism, in the revised version of the manuscript we added line about the limitations of our MD approach in the methodology section.

Issue 6)

The manuscript also emphasizes the RMSD of 1-1.6Å between the corresponding Access, Binding and Extrusion states of ArcB and MdtF, seeming to emphasise that these are large transitions. In fact, these RMSDs show that the protein structures are extremely similar. Even RMSDs of 3 Å are within the expected conformational spread of the cryoEM structures.

The RMSD differences between protomers was deemed to be significantly similar if <1.5 Å based on our analysis and previous analysis by others in the field (Ornik-Cha, A., Wilhelm, J., Kobyłka, J. *et al.* Structural and functional analysis of the promiscuous AcrB and AdeB efflux pumps suggests different drug binding mechanisms. *Nat Commun* 12, 6919 (2021). <https://doi.org/10.1038/s41467-021-27146-2>). The relatively small RMSD is a recognition that these RND transporters have large areas of conformational similarity due to the limited homotrimeric topography within a membrane. Therefore, global RMSD analysis cannot suitably isolate where significant changes are occurring locally within the structures across homologs and protomers. However, detailed analysis of localised changes to porter entrances (as seen in AcrD for example - Zhang Z, Morgan CE, Cui M, Yu EW. 2023. Cryo-EM Structures of AcrD Illuminate a Mechanism for Capturing Aminoglycosides from Its Central Cavity. *mBio* 14:e03383-22. <https://doi.org/10.1128/mbio.03383-22>

) or transmembrane conformations (as presented here), can be significantly consequential to the functional rotation mechanisms of RND transporters, as has been defined throughout the literature. We agree that we did not present sufficiently detailed analysis in our previous manuscript but have now provided further evidence of this distinct protomer state in MdtF, which would create a unique functional rotational mechanism. We detail the extra analysis performed in our response to key point 1 earlier.

The authors also state on Page 13 “although our MD simulations reveal relatively large structural departures in the MdtF dynamics compared to the cryo-EM geometries, the simulations of AcrB provided findings that are in good agreement with experimental data³⁴”.

There is no evidence presented to show that the MD simulations have relatively large structural departures in the dynamics compared to the cryoEM geometries. This would require some kind of measurement of the conformational dynamics, etc, RMSD, RMSF, cluster analysis, etc. No RMSDs or other measure of global conformational similarity or difference has been presented. The authors also do not explicitly note that MD simulations are performed in a membrane that’s very different from the lipidome identified.

We thank the reviewer for pointing out this unclear statement. The sentence on Page 13 specifically referred to the PC1–PC2 cleft, where notable deviations from the initial cryo-EM (or crystal) structure were observed (Figure 4A,B). To better clarify this point, we have also performed C α RMSD analysis as a measure of the overall AcrB and MdtF departure from their reference experimental structures during the molecular dynamics simulations. The results, reported in Supplementary Figure 22, demonstrate that on a global level the systems are very stable, with consistent remaining RMSD values throughout the simulation. We added a comment on this result in the revised manuscript.

The authors describe transmembrane conformational changes and acid responsive transport mechanism not found in other RND efflux pumps, which they have identified from the cryoEM structure, MD simulations and functional assays. The key questions here are whether these are novel, whether the conclusions are convincing, and how the MD simulations and functional assays were performed.

The authors state that MdtF gives moderate efflux under neutral conditions. They speculate this may be related to maintenance of PMF under stress conditions. It is not clear why these ideas are linked? The rationale for this speculation needs to be set out more clearly.

We agree that these discussions were unclear. We have updated the discussion to more clearly expand on the conclusions, detailing the potential rationale for the observed moderate MdtF activity within physiological conditions and its acid-activated efflux activity. Bringing in detail on what is known on the upregulation of *mdtEF* in the literature and how the functional characteristic of MdtF could be advantageous as part of a response to exhausted conditions (which create pressures for strong PMF maintenance) that presage acid-shock.

The authors go on to state that under acidic stress, there is increased efflux efficiency to “battle xenobiotic attack and host-toxin clearance”. What is meant by efficiency? Energy efficiency? Increased activity?

We have changed the language here to refer to the activity of efflux rather than the efficiency of efflux.

Issue 7)

The authors over-claim their results without substantiating them. The crux of the authors findings (as it is currently written) is that they have cryoEM structures for MdtF, and they show that when MdtF is in a membrane that more closely represents a non-stressed E coli membrane, its conformational dynamics are slightly differently to AcrB. The manuscript basically says that MdtF is more rigid in the core and the peripheral regions of the protein are floppier.

We have updated the manuscript text to better describe the structural and dynamical differences found between MdtF and AcrB. We also clarify any misunderstandings and ambiguous language surrounding the capture of MdtF in a membrane; being more explicit that MdtF was captured in a SMALP from stationary phase membrane fractions and that future work will need to be done to understand how alternative membrane compositions influence the structure-function of MdtF and AcrB.

The authors make very big claims about the distributions of point mutations for the 30% of the sequence that isn't conserved between AcrB and MdtF. The authors support this with Figure S10, which shows that mutations are spread throughout the protein. However, this comes across as though the authors are trying to re-invent 50 years of ‘homologous proteins share the same fold’, and this is also the case for every other globin, ion channel, membrane transport protein and enzyme that is known to biology. Likewise, while it is important to point out that AlphaFold3 cannot build a machine learning model of any RND efflux pump in a trimeric A-B-E conformation, this is a known limitation of AlphaFold ML/AI models. This is an important point that should be stated.

Please point it out, but cite all the other people who have also pointed this out! This should contribute to the body of literature, rather than fragmenting it. “Our study provides the first mechanistic view of how MdtF can recognise a wide range of harmful cellular and xenobiotic substances, incl. antibiotics”. The authors also state that this functional data will help with EPI design. I’m not sure I agree with that claim, but I think presenting the mechanistic basis of an altered function point mutation is worthwhile on its own merits.

We thank the reviewer for the points raised, we have since updated the manuscript to remove statements relating to sequence distribution and fold as they are redundant, as pointed out. Following this, we have also updated our references to acknowledge work that has previously looked at AF limitations in looking at functional asymmetry in homomers.

The discussion has also been updated to more clearly delineate the implications of this study in EPI design as opposed to being a direct basis for the design of novel therapeutics.

Issue 8)

The figures generally need work.

Several of the figures in the main text and supplementary have been updated to provide better dissemination of the findings.

Figure 1A and Figure 3 are generally unclear and do not support the written rationale adequately. The Position of V610 /F610 not clear from the simulations. In Figure 3A – you can’t see where V610 is and where F610 is. This is important! The image with the hydrophobic nook has the larger side chain and the hydrophobic nook is not apparent in the figure. Residues should be coloured differently in part A to highlight them. Even the position of these residues is unclear from the figure. I would have thought that V610 was part of the switch loop. If it is, please state this explicitly.

A new figure highlighting the location of the V610 residue has been included in Figure 3 and has been described in the main text of the manuscript.

Figure 3B – fluorescence polarisation is not explained in the main text or the methods. This needs to be clarified for a non-expert audience.

An explanation of the fluorescence polarisation assay has been added to the manuscript to clarify its use in this investigation.

Figure 2C – Supplementary Figure 25 is so much easier to understand than Fig 2C because of the use of transparency in S Fig 25.

This has been corrected.

Figure 4 needs further explanation. The significance of A675, S676 is not explained adequately in either the figure caption or text.

A675 and S676 are now labelled (in brown colour) in Figure 4 and we have added an expanded explanation of Figure 4 in the text to better describe the experiment and what was measured.

Figure 5A doesn't actually show you where the proton relay switch is, however the top view of Figure 5C is nice. Figure 5D gives the H-bond occupancy, however this is not a standard format for a H-bond occupancy plot and it's not stated in the text or figure legend what the box plots mean! These are not well explained and they are not presented in a standard format that is easy to interpret.

We have included a more zoomed in version of the proton relay network within Figure 5A to show the location of the proton relay network residues within the transmembrane domain of MdtF. In addition, we have modified Fig 5D with a bar plot to make it clearer.

Gels and micrographs in figure S1 poor image quality, low resolution, jpeg artefacts and fringing, however we acknowledge this could that be a problem with the proofs. Figures S11 S12 have inadequate labelling. Can't tell the degree to which visual differences between protomers in S11 are due to angle. Pairwise RMSD would likely support the author's point.

We have added pairwise RMSD analysis to these figures. The image quality noted is because of proof artefacts.

Supp Fig 12 states that AF3 was not successful, however the models look the same by visual inspection. Also the colour key is not given, so you can't tell which is which. Again, a pairwise RMSD would show the AF prediction monomers are the same while the structural monomers are distinct, which would further support the authors' important point about the importance of structural data over AlphaFold predictions. While this is done for AcrB and MdtF in Figure 2B, it would be great to have it for the AF predictions as well.

This analysis has now been included in the manuscript in Supplementary Figure 11.

Supp Fig 26 – the whole argument till now is about the binding pocket, now we're switching to channel 1 channel 2 and channel 3... Please link these clearly in the main text if it's important. Supp Table 3 – docking free energies. How reliable are these? Connection back to the general idea of how this works together with everything else is not clear.

We have now provided a clearer link to the channel 1-3 assessments and associated figures and supporting discussion in the main text. The supporting discussion has also been modified to improve its readability.

Regarding docking energies, we used the Glide scoring function, which is well-validated for protein–ligand docking (see e.g. [10.1039/c6cp01555g](https://doi.org/10.1039/c6cp01555g)), to identify most likely binding poses at different regions of the transporters. While docking scores are empirical and not absolute binding free energies, they are useful for comparing relative binding trends. Our aim was to assess differences between WT and V610F, and between CH2 and CH3 regions, using ensemble structures and extensive ligand sampling to provide qualitative insights into ligand recognition in MdtF variants.

We have added a sentence in the Methods section: “*Despite their widespread applications, docking methods are known to have inherent limitations in accurately predicting absolute binding affinities, and are best interpreted for relative ranking and qualitative comparisons across ligands and receptor variants [10.1128/aac.00496-20, 10.1021/jm050362n].*”

Issue 9)

Why were the simulations done in KCl?

How were protonation states assigned? Were pKa values for residues predicted with some tool such as a PropKa, or were protonation states assigned based on canonical pKa values? The methods state that residues were protonated at physiological pH, but do not indicate the physiological pH relevant to their system.

Thank you for your insightful comments regarding the simulation setup. We have addressed your concerns under the following points:

Choice of KCl in Simulations:

To mimic physiological ionic conditions, we used a 0.15 M KCl solution as the buffer electrolyte in our simulations. This simple bi-ionic solution was chosen to both limit finite-size effects on the membrane and provide K⁺ and Cl⁻ counterions to neutralize the system's net charge, as required by the Particle Mesh Ewald (PME) method for

treating long-range electrostatics [Dickson et al., 2022, *J. Chem. Theory Comput.*, 18, 1827–1843, DOI: 10.1021/acs.jctc.1c01217]. The choice of KCl was motivated by both biological and computational considerations:

1) K^+ is present at higher concentrations intracellularly (10.1016/j.jmb.2021.166968, 10.1128/IAI.00766-20) compared to Na^+ , making KCl a more physiologically relevant electrolyte for bacterial physiology and efflux [Lodish et al., *Molecular Cell Biology*, 6th ed.; W. H. Freeman: New York, 2007].

2) Amber Lipid21 force field validation studies have shown that bilayers can undergo condensation in the presence of ions (K^+ , Na^+ , Ca^{2+}) leading to an underestimated area per lipid compared to experimental values. Among these, K^+ exhibits only weak binding to lipid headgroups, better reproducing realistic membrane properties and minimizing the risk of bilayer condensation or complex dehydration [10.1021/acs.jpcc.9b06091]. This behaviour, which is related to challenges in reproducing the correct ion-lipid pairing, has been tested by several approaches such as deriving pair-specific ion parameters (for sodium; see 10.1039/C6CP04883H), modifying the van der Waals interaction (see 10.1039/C7CP08185E), or adjusting the lipid model (see 10.1021/acs.jpcc.7b12510). A similar effect was also noted for the GROMOS forcefield where K^+ ions interact less strongly than Na^+ with lipid headgroups, reducing membrane perturbations and potential simulation artifacts [10.1529/biophysj.107.116335].

Physiological pH and Protonation States:

While acidic conditions can induce overexpression of MdtF, our simulations focus on the functional mechanisms and conformational dynamics under physiological conditions, avoiding the complexity of membrane polarization and altered protonation states at low pH. This choice aligns with experimental observations that MdtF is expressed and functional near neutral pH [10.1128/AAC.01735-20; 10.1128/AAC.01803-20].

Protonation State Assignment:

Protonation states of ionizable residues were assigned based on pKa predictions obtained using PROPKA3 [10.1021/ct100578z; 10.1021/ct200133y]. This empirical tool calculates residue-specific pKa values considering the local environment, enabling accurate assignment of protonation states at physiological pH.

We believe this setup provides a reliable and biologically relevant framework for investigating ligand binding and transporter dynamics in MdtF.

Reviewer #2 (Remarks to the Author):

This manuscript describes the first cryo-EM structure of the RND-type efflux pump MtdEF from *E. coli* that is expressed during early stationary phase growth and in response to environmental stress, such as low pH. The authors provide interesting evidence for significant differences in the tertiary structure of MtdF, as compared to AcrB, which reflect its distinct role during stationary phase and as a stress response mechanism. The structural data is augmented by MD simulations and wet lab data. For the most part, the data support the conclusions of the authors. Since MtdF has not been as extensively studied as the constitutively-expressed AcrAB pump, the manuscript would provide a valuable contribution to the field. However, there are several issues that need to be addressed before publication.

General comments.

Since the description of the cryo-EM structures in the first half of the results sections contains detailed descriptions of the three functional conformations of the MtdF monomers and their conformation transitions, it would be helpful for the reader who is not an expert on RND-type pumps to include a paragraph in the Intro needs more background information on RND structure and function.

This has been addressed in the introduction to better explain the functional asymmetry and its related mechanism.

The Introduction and the subsections of the Results describing the cryo-EM results (including the figure legends) and discussion would benefit from extensive editing for grammar, clarity, and accuracy. Some of the descriptions of the key findings are cursory, and would be difficult for a non-expert to follow.

This has been improved.

Most of the data cited in the main manuscript is in the supplemental data section (31 figures and 4 tables). Some of these data are important for supporting the conclusions of the paper. It was tedious to have to refer to the supplemental data 34 times (one of the tables was not mentioned in the text). Most readers will not bother to look at the supplemental data. I suggest that the authors reevaluate the data and move the figures that are key to supporting the main conclusions to the main paper.

Supplementary figures have been condensed and improved, and we have now provided a table of contents for better navigation of the Supporting Information and brought some supplementary information into main figures, where we thought it

best aided the understanding of the findings, as suggested. Although we do note that there is still a lot of supplementary data included but we believe the data provided is all necessary to provide confidence of the structural and dynamical analyses that support the main findings.

The axis labels on the supplemental figures that show the results of laboratory experiments are too small to read. Please increase the size of the figure and font size.

This has been corrected.

Some specific comments related to the general comments.

Page 7. The authors use the phrase “promiscuous substrate efflux” to describe that activity of MdtF. However, the term “polyspecific binding or efflux” site provides a more accurate description, as the binding site accommodates substrates with a set of general chemical properties but does not bind to everything.

This has been corrected.

Page 8. The FP data indicates that MdtF binds to Rhodamine. Therefore, it is curious that the authors were unable to obtain a cryo-EM structure of Rhodamine bound to WT MdtF. The authors should address this discrepancy. Is it possible that the FP assay is not measuring specific binding to the substrate binding site?

The FP assay revealed a similar binding affinity of R6G to WT and V610F, but displayed different polarisation signal amplitude (lower polarisation in WT compared to V610F). We attribute this to a more heterogeneous population of ligand binding modes in the WT versus V610F which then led to poorer density being able to be achieved and left the ligand unresolved. Whereas the more restricted and homogenous binding in V610F enabled the visualisation of unambiguous density to be achieved for the R6G substrate. This detail has been reworded from the perspective of the WT to be clearer and demonstrate the results likely indicate that the binding is heterogeneous in WT MdtF and therefore not identifiable.

The authors state that “Surprisingly, we found that the exponentially growing and the stationary cells producing MdtEF differ in the accumulation of HT with the stationary cells accumulating significantly lower amount of HT than the exponential cells (Fig. 6A and Supplementary Fig. 23A).” This data is in Supp Fig, 23B. However, the difference between experiments shown in 6A and supp fig. 23B is not clear. They are described as essentially the same experiment, but the results are very different.

The figure, figure legend, and relating text in the manuscript has been updated to better explain the finding which make the results clearer.

Supp Fig 23A, which needs labels for each protein, shows pump expression is the same in exponential and stationary cells. However, the lack of expression of the pore in stationary phase leaves the question of altered membrane permeability unanswered. The observation that HT accumulates to the same level in exponentially growing and the stationary cells with vector only (even without pore), suggesting that permeability of HT is the same in exponential and stationary phase cells. Therefore, decreased HT accumulation in stationary phase cells could be caused by increased pump activity in stationary phase. Since membrane composition markedly changes in stationary phase, it seems possible that the activity of MtdF, an integral membrane protein, may be responsive to the altered membrane composition.

We considered this insightful analysis carefully. However, in Supp Fig 23A (now Supp Fig 27A) the Pore expression between stationary phase vector-only and MdtF pump conditions shows a significant decrease between each condition. With a much lower level of Pore expressed in the MdtF pump condition. This could be the reason for the reduced accumulation observed in this condition versus the vector only, and not improved efflux activity of MdtF in the stationary phase. Because of this we still support that these experiments are inclusive and restrain from forming conclusions which may be misconstrued.

We do believe that understanding RND efflux activities across exponential and stationary phases for a range of transporters would be insightful and suggest in our discussion that future work in this area would be revealing. However, it would require careful and extensive analysis which is outside the scope of this study.

Fig 6B. As a control, It would have been useful to include AcrAB in this experiment.

This was previously performed and published, and the paper has been referenced. This has now been better stated within the manuscript, as we agree this is an important point.

Figs. 1A and 6C. The figures shows that RND substrates are extruded from cytoplasm. Since the substrate-binding site of MtdF is in the periplasmic space, substrates are extruded from the periplasmic space. I suggest editing the figures accordingly.

These have been updated.

The MIC data shown in Supp table 4 is not mentioned in the text. In addition, the table does not contain susceptibility data no MICs for the V610F mutant. While this data exists in the literature, it would be useful to include MIC data for a broad range of substrates in this manuscript to verify the phenotype of the WT and V610F mutant pumps using the engineered strains described in the manuscript.

MIC data relating to the V610F mutant has been collected and is now included in the manuscript along with a description of the results.

Page 16. Please note that the sentence containing “such as the ESKAPE (Enterococcus faecium, Staphylococcus aureus, Klebsiella pneumoniae, Acinetobacter baumannii, Pseudomonas aeruginosa, and Enterobacter spp.) classes” should be revised, as the Gram-positive pathogens in this list do not encode RND-type pumps.

This has since been revised and removed from the manuscript.

Reviewer #3 (Remarks to the Author):

The manuscript submitted by Lawrence and colleagues describes the molecular characterisation of MdtEF, a proton motive force-driven efflux pump from the resistance-nodulation-cell division superfamily. The manuscript is overall well-written, is easy to understand and is innovative in the findings presented.

I have reviewed the lipidomics aspects of the manuscript. The authors should consider the following feedback.

1. The GC-MS and LC-MS raw and processed data should be made publicly available in a metabolomics data repository (Metabolights or Metabolomics Workbench). The output data should also be included in supplementary information for readers to review.

This has been uploaded and the supplementary information has now been included.

2. The GC-MS/FAMES extraction method is appropriate. The GC-MS assay has a short analysis time which would not support full chromatographic resolution of all fatty acids. The authors need to provide evidence that all fatty acids reported are identified correctly with high confidence. A commercial mixture of bacterial acid methyl esters was used for identification of fatty acids but no information on the source or composition of this mixture was provided and should be provided. Also, were all detected FAMES from the sample present in the commercial mixture including cyclopropanated fatty acids or were some detected fatty acids in the samples not present in the commercial mixture? If the latter then how were those fatty acids detected but not in the commercial mixture identified? Full methods are required.

All fatty acids detected were present in the commercial mixture and the source and composition of the mixture has now been detailed in the GC-MS data figure legend. Chromatograms have been included in the supplementary information to acknowledge this.

3. GC-MS/FAMES assay. Can the distribution of carbon number be reported as this will also be of biological relevance/importance.

This has now been added to the GC-MS figure in the Supplementary.

4. The LC-MS assay appears to be appropriate. However, the annotation/identification of lipids is not performed appropriately. Identification has been performed using accurate mass only which provides probable errors. Why were MS/MS data not collected and used in the identification process? Many of the lipids reported have a

mass which can also be linked to another lipid (e.g. PC and PE species as well as PG and PS species). Without further inclusion of methods applied in Lipidmaps including m/z error, adduct type etc and further experimental evidence, lipid identifications should not be reported.

We performed MS/MS data collection on our lipid extracts and confirmed the identity of the most abundant PE and PG lipids in our sample. Confirming that PE is the dominant lipid. However, we did not achieve sufficient MS/MS quality for all lipids identified and see these remaining ones were defined by accurate mass alone (detail on the decision process now expanded in the Methods section). When doing so we excluded any potential phosphatidylcholine (PC) matches as *E. coli* doesn't naturally produce PC phospholipids.

5. LC-MS assay. The parameters used by the software and the strategy applied for processing of the data should be reported. Were data processed to search for specific m/z values in a targeted way or was an untargeted strategy applied? Full information should be included.

We have included more detail on the data processing approach in the Methods section. In this work we had not set out to perform a lipidomic analysis of the bacterial membrane and the SMALP discs – this would require further workflow and data analysis optimization - and the intention was to confirm the identity of the predominant lipid species in the SMALP samples for ligand-fitting choice to the phospholipid density observed in the cryo-EM map. The LC-MS data confirmed that PE was dominant and so this was selected for map fitting – indeed PE is known to be the dominant phospholipid in *E. coli*. To reflect the limited nature of our analysis, we have now removed reference to 'lipidomic' analyses in the text and instead referred to LC-MS analysis of extracted lipids for improved clarity of the approach taken.

Reviewer #4 (Remarks to the Author):

We thank you for your review.

Reviewer #5 (Remarks to the Author):

Escherichia coli multidrug transporter MdtF is a special resistance-nodulation-cell division (RND) transporter because it is upregulated and functional in low oxygen, extreme acid and nutrient starvation conditions. In this manuscript, Lawrence et al. solved the cryo-EM structures of Apo-MdtF WT, Apo-MdtF V610F and R6G-MdtF V610F within native-lipid nanodiscs. The thorough comparison of the asymmetric structures of MdtF and its close homolog AcrB, in addition to functional assays and substantial computational analysis such as molecular dynamics simulations and docking, reveals the structural and functional specialties of MdtF. However, the linkage between structural findings and the pump function particularly in extreme conditions is somewhat insufficient.

There are a few major concerns to be addressed:

1. The structural analysis mainly focused on Apo-MdtF WT, which is overall alike to AcrB. The structures of Apo-MdtF V610F and R6G-MdtF V610F were not described or compared in enough details. Are these two structures also asymmetric and alike?

Yes, the MdtF V610F structures are also similarly asymmetric. This point has been updated in the text, and figures relating to V610F asymmetry have been included in Supplementary Fig. 19.

It is confusing since the author stated that “MD simulations reveal that the V610F mutation induces ..., resulting in a somewhat more symmetric MdtF conformation ... (P11)”. Why was only one R6G ligand identified in the substrate-bound V610F structure (9QPT)? In which protomer of the R6G-MdtF V610F structure was the bound R6G identified? For the analysis and MD, it was often unspecified which chain of Apo-MdtF V610F or R6G-MdtF V610F was used.

Because the volume of the pockets is large, it is common to resolve substrates only within the one state of RND homotrimeric transporters [https://doi.org/10.1038/nature05076]. Within the binding state certain moieties are more likely to be resolved as the porter-domain distal binding pocket within this conformer is more conformationally restricted. R6G was found in the conventionally named ‘binding’ state of MdtF V610F and reference to this in light of the known literature has now been implemented into the manuscript.

Also placed in the introduction is a better explanation of each of the states of the protein which will re-iterate the functional purpose of these pumps and each of the

protomers. For MD, we have performed analysis for each chain individually and have now clarified this more explicitly in the revised manuscript.

2. As a major anaerobic RND transporter, nitrosyl indole derivatives are the important substrates for MdtF. The experimental structure of MdtF in complex with nitrosyl indole derivatives is more meaningful. Why was only R6G used as the ligand in the complex structure determination? The binding affinity estimation of MdtF WT and V610F by docking (Supplementary table 3) suggested that the CH3 binding might be lightly stronger for nitrosyl indole than the DBP binding. It is possible that the nitrosyl indole derivatives are favored by the different binding site in MdtF compared to R6G in MdtF or AcrB. In addition, it is more convincing to provide experimentally determined binding affinities of MdtF.

Nitrosyl indole is natural to MdtF, and we agree that a structure in complex with this substrate would be very meaningful, but it cannot be purchased and needs to be synthesised through non-routine synthesis outside the scope of the labs, whereas R6G is commercially available. R6G was also chosen because MdtF is known to confer resistance to it and the V610F mutant had altered efflux to this substrate; the effect of V610F on nitrosyl indole efflux has not been determined. R6G also has intrinsic fluorescence that could be exploited for biophysical assays, including FP assay which allowed us to calculate a Kd value. We agree that it is possible that the nitrosyl indole derivatives are favoured by the binding sites provided in MdtF compared to AcrB, and other substrates. We now mention this possibility in the discussion of the manuscript but as we are cautious to use the docking data alone to suggest nitrosyl indole translocation mechanisms.

3. All the cryoEM structures, MD simulations and most of functional assays were conducted at physiological pH. However, MdtF is known as an acid-responsive transporter in some extreme conditions. The identified proton relay residues of MdtF were highly conserved among RNDs. Thus, these results didn't provide sufficient evidences for the acid-responsive specialty of MdtF. In addition, from the NPN efflux assay, it was shown MdtEF confers enhanced activity at lower pH. This result is intriguing. Identifying the key residues contributing to this unique property of MdtF would help to elucidate the special acid-responsive mechanism that may distinguish MdtF from AcrB and many other RNDs.

We thank the reviewer for raising this important limitation of the study. We have now included a section on this in the discussion describing it and about how this is the next step to take now we have the first structures of MdtF. We also describe how this would need some technical rethought however, as the SMA polymer is susceptible to low pH and becomes insoluble around pH 5 and alternative acid-

tolerant co-polymers would need to be explored to capture MdtF in a native lipid nanodisc. We believe our results can shed significant light on understanding the structure-function of MdtF, especially in comparison to the well-characterised AcrB, whose cryo-EM structures in SMALPs were also collected at physiological pH.

Some minor comments are listed below that may improve the clarity and readability of the article.

1. In the Fig 3, location of the V610F mutation site and DBP in the porter domain is obscure. The protein surface (panel A) could be colored according to the hydrophobicity scheme.

This has been included.

2. The Supplementary Fig 9 could be labeled or marked with secondary structural elements, domains and essential residues.

This has been included.

3. Related to Supplementary Fig 12, although it was stated the asymmetric structure of MdtF WT was not captured by AlphaFold 3 prediction, it is not clear which state of the protomers the AF predicted monomer (or trimer) resembles.

Labels delineating each of the protomers has now been included and the figure has been updated.

4. The assignment of each monomer state of MdtF WT was based on the RMSD calculations of the structural comparison with AcrB. It is not clear if the only C alpha atoms were counted in the calculation for each monomer. As to the Chain B of MdtF WT, the RMSD values for “Access” (1.6 Å and “Binding” (1.9 Å) are very close but obviously less significant than the best RMSD for the other two chains (Chain A, 1.0 Å; Chain C, 1.1 Å). The authors claimed Chain B as the “binding-like access R2” state and reported some major TM movement of this “Access” state. It is not clear how similar their porter domains are when compared with AcrB. Regional comparison and RMSD calculation may help to explain what structural variance may cause the “high” RMSDs.

We have now provided further, localised C α -backbone RMSD analysis in Figure 2 and associated supporting information figures add Supplementary Table 7. We also compare MdtF structures to AcrB structural models in both detergent micelles and SMALPs. This new analysis helps to better portray the significant differences

between MdtF and AcrB asymmetric states, especially in the transmembrane domains of the access state. We thank the reviewer for this suggested analysis; this has significantly improved the manuscript and confidence in the conclusions.

5. In the Supplementary Fig 13A, it is not clear what state of protomers was presented.

The related figure was a representative, however, has since become redundant and removed with the new edits as the hoisting and switch loops have since been highlighted in other figures.

6. P11, “there were more distinct intermediate and extreme conformations allowed by (3 vs. 2 and 2 vs. 1 in the access and binding states, respectively, Fig. 4B”, what exactly do “3 vs. 2 and 2 vs. 1” refer to?

The statement “3 vs. 2 and 2 vs. 1 in the access and binding states, respectively” refers to the number of peaks corresponding to distinct conformational populations observed in the pseudocontact (PC1–PC2) distribution plots shown (Fig. 4B). We have clarified this in the revised text.

7. In the Supplementary Fig 20, the contour levels for the map densities of the proton relay residues were not specified. Please also note the cryoEM map is not calculated based on electron densities.

This has been updated.

8. In the Supplementary Fig 28-31, it is better to show the site of V610F to mark the main structural difference between WT and V610F. The docking scores or energies should be provided in addition.

We have now highlighted the mutation site V610 in the DBP docking poses (now found in Supplementary Figure. 15) to visually indicate the key structural difference between WT and V610F. Additionally, we have included the docking scores for all poses of DBP, including those docked in channels 1, 2, and 3 (now found in Supplementary Fig. 16-18).

9. There are numerous typos in the supplementary figures, eg, Fig S6, S17, S20, S22, S24, etc.

These have been corrected.

10. The blotting figures are too small in the Supplementary Fig 21, 26C.

These have been improved and uncropped versions of the blots have been included.

Reviewer #6 (Remarks to the Author):

Laurence and colleagues report on the structure-function relationship of MdtEF, a RND multidrug transporter overexpressed under anaerobic or stress conditions. To understand the particularity of MdtF transporter, they decided to compare it with the constitutively expressed transporter AcrB. The methodology developed in the ms is based on a structural method, cryoEM combined with molecular dynamics, and functional assays. Consistent structural and functional results were obtained providing clue to better understand how MdtF functions.

General comments

Solid work has been carried out to explore the molecular basis of MdtF function. On one side, the structure of MdtF wt and mutant with and without ligand were determined at neutral pH, with native lipids using the SMA polymer. MdtF wt exhibits structural differences from AcrB. Mutation experiments in binding site and MD simulation seem to suggest that MdtF activity can be modulated by changing the binding site accessibility. On the other side, functional analysis indicates that external pH affects MdtEF activity. Its efficacy is increased at low pH. This raises an important question. What is the conformation of MdtF at low pH? Is there a change for the binding site? This piece of the puzzle is missing to fully understand the role of pH in the functioning of MdtF.

We thank the reviewer for raising an excellent point. This is an important limitation of our study and we have now included a section on this in the discussion, focused on how this is the next step to take now we have the first structures of MdtF.

The authors proposed that the MdtF identity is defined by its proton relay that could be a “truly distinct mechanism of coupling between proton translocation and substrate export in MdtF” (p13). It is not clear how to correlate the observation in the computational experiment that R969 is directly involved in the proton relay and why this would have an impact on substrate export. This point needs to be clarified. Moreover, it is not clear how the proton relay would interplay with the pH-dependent activity (in the paragraph “MdtF drug transport is responsive to proton load”).

RND efflux mechanism and its relation to the proton relay has been expanded in text in the introduction to provide clarification on this point.

Indeed, it is important for the reader to have a clear idea of what makes MtdF specific: is it specific substrates that are produced under extreme conditions that need to be expelled or is it the conditions themselves that enable the function of MdtF?

This is a very good question which we do not have a direct answer to but makes for an interesting discussion around our findings and their limitations. We have now enhanced our discussion by using the point raised to frame parts of it.

Experiments with the V601F mutant suggest that it is possible to change substrate specificity. There is a slight increase in the affinity of R6G, making it possible to resolve the structure of the R6G-MdfF complex (V601F). This mutation appears to confer a function similar to that of AcrB at neutral pH. It would be interesting to assess whether the activity of MdfF (V601F) is also pH-dependent like MdtF wt.

We agree this would be interesting to perform and could form part of future work investigating pH effects on MdtF structure-function.

Molecular basis of MdtF efflux has been studied at low pH and not under anaerobic conditions, I would suggest changing the title of the ms accordingly

We have further clarified that MdtF upregulation is distinctly associated with anaerobic conditions in *E. coli* amongst its 20 efflux related genes, defining it as the anaerobic-associated RND transporter. Induction of *gadE-mdtEF* is thought to be rigged under many different circumstances, such as stationary-phase and anaerobic transitions, that presage – instead of reacting to - an encounter with extreme acid, where *E. coli* cannot grow. We believe we observe this phenomenon happening at the molecular level of MdtF (acid-sensitive) transport action. However, we take the point that MdtF is not solely associated with anaerobic conditions and so have altered the manuscript title (to: “Molecular basis for multidrug efflux by an anaerobic-associated RND transporter”) to better express this point.

Additional comment

Figures 1A and 6C appear redundant. I suggest to group them in one figure..

This has been updated.

Reviewer #1

The authors have revised their manuscript extensively and have addressed the majority of the reviewer comments. I have two minor comments that should be addressed before publication:

1) Figure 5D: what do the circles above and below the bars on the bar graph mean? Presumably they represent errors, but this should be explicitly stated.

The circles are indicative of an individual data point which represents the average hydrogen bond occupancy over each of the three repeated simulations. To increase clarity, this has been better explained within the figure legend.

2) There are still a few grammatical and typographic errors that need to be corrected.

The grammatical and typographic errors have been corrected.

Reviewer #3

The authors have provided a revised manuscript based on reviewer's feedback. For reviewer 3 some of the feedback has been applied appropriately in the revised manuscript though other feedback has not. The authors need to consider the following.

1. The GC-MS data have been included in a metabolomics data repository (Metabolomics Workbench). However, the LC-MS data has not been included in a data repository but should be.

LC-MS data has been uploaded to the MetaboLights database (under accession code: MTBLS13050) and this has been included in the Data Availability statement.

2. The authors have included a GC-MS chromatogram of the BAME mixture and the sample in supplementary information which is appropriate. However, my feedback for version 1 of the manuscript was to define with confidence the fatty acid methyl ester identifications. This has not been performed. For example the BAME mixture has 26 components but the BAME chromatogram has only 21 peaks, where are the peaks for the other 5 BAMEs? I suspect there is co-elution. Also, how do the authors know which BAME relates to which peak with a specific RT in a mixture of BAMEs, surely this would require analysis of each BAME separately? Further information is required.

Each of the 26 individual peaks from the standard mixture were identified by GC-MS, however, some were recorded with a lower intensity. The area under the curve for each peak have since been recorded and added to the supplementary information to highlight this disparity. Nevertheless, each of the individual peaks of the standard sample that were used to relatively quantify the percentage abundance for each lipid type within the SMALP-extracted sample were of higher intensity and could easily be identified. The retention time relating to the sequential separation of the 26 fatty acyl chains based on their size has been characterised for the standard sample by the manufacturer (1). Therefore, the separation of 26 individual peaks within the standard sample for our experiments provides confidence that the acyl chains within the standard sample were identified and each acyl chain within the SMALP-extracted sample could be identified based on these specific retention times as performed previously (2). Nonetheless, conclusions relating to this data are minimal in which the experiment was used to characterise relative fatty acyl chain types and the relative proportion of each grouped class and not to identify each individual fatty acyl chain present.

3. The part number of the BAME mixture should be included in the methods section.

The part number (47080-U) and Lot number (LRAD6478) of the BAME mixture (Sigma Aldrich) from the manufacturer has been included in the Methods section.

4. The LC-MS data processing software parameters should be included as recommended previously.

The LC-MS data processing software parameters have been included within the Methods section of the manuscript and in the MetaboLights repository submission.

References

- 1 Merck. (2022). *Bacterial Acid Methyl Esters CP Mixture Certificate of Analysis*. Retrieved from: https://www.sigmaaldrich.com/certificates/Graphics/COfAInfo/fluka/pdf/rtc/47080-U_LRAD6478.pdf
- 2 Berezhnoy NV, Cazenave-Gassiot A, Gao L, Foo JC, Ji S, Regina VR, Yap PKP, Wenk MR, Kjelleberg S, Seviour TW, Hinks J. Transient Complexity of E. coli Lipidome Is Explained by Fatty Acyl Synthesis and Cyclopropanation. *Metabolites*. 2022 Aug 24;12(9):784. doi: 10.3390/metabo12090784. PMID: 36144187; PMCID: PMC9500627.